# Dual function of a highly conserved bacteriophage tail completion protein essential for bacteriophage infectivity
Isabelle Auzat [1] ✉, Malika Ouldali [2], Eric Jacquet [3], Beatrix Fauler[4], Thorsten Mielke[4] & Paulo Tavares [1]

Infection of bacteria by phages is a complex multi-step process that includes specific recognition of the host cell, creation of a temporary breach in the host envelope, and ejection of viral DNA into the bacterial cytoplasm. These steps must be perfectly regulated to ensure efficient infection. Here we report the dual function of the tail completion protein gp16.1 of bacteriophage SPP1. First, gp16.1 has an auxiliary role in assembly of the tail interface that binds to the capsid connector. Second, gp16.1 is necessary to ensure correct routing of phage DNA to the bacterial cytoplasm. Viral particles assembled without gp16.1 are indistinguishable from wild-type virions and eject DNA normally in vitro. However, they release their DNA to the extracellular space upon interaction with the host bacterium. The study shows that a highly conserved tail completion protein has distinct functions at two essential steps of the virus life cycle in long-tailed phages.

Tailed bacteriophages (class *Caudoviricetes*[1]) are the most abundant viruses that infect bacteria, having a tremendous impact on bacterial communities' dynamics and evolution[2–4]. Understanding how their viral particles are built and how they successfully deliver the phage genome to host cells is of central importance to develop strategies to target phage infection. Tailed phages are composed of a capsid (or head), which contains and protects the phage genetic material, and of tail. The tail is responsible for specific recognition and attachment to the surface of the target cell, followed by transfer of phage DNA into the bacterial cytoplasm.

Viral particles of tailed phages can be distinguished according to their tail morphology: short (podoviruses), long and contractile (myoviruses) or non-contractile (siphoviruses)[5]. Short tails of podoviruses assemble directly at a specialized portal vertex of the capsid, whereas long tails are constructed in an assembly pathway independent from the capsid[5]. Long tail assembly starts at the end of the tail distal from the phage capsid by formation of the initiator complex that serves as the adsorption device to the bacterium. Next, the tail tube protein (TTP) polymerizes around the tape measure protein (TMP) to form the helical part of the tail. An additional sheath surrounds the myovirus tubes. The sheath structure contracts at the beginning of infection, before DNA ejection. Finally, another set of proteins taper the tail tube[6,7]. They build an interface for binding to the connector structure found

at the capsid portal vertex[8,9]. The complex formed by the connector and the tail tapering proteins is named the phage particle neck. Although the function of connector components is well-studied[8–10], the assembly and function of tail proteins at the tail-to-head interface are poorly understood. Their investigation is of critical importance to understand the mechanisms how the tail-to-head interface builds-up and its function for successful transfer of viral DNA from the phage capsid to the host cell cytoplasm.

The best characterized superfamily of tail proteins within the tail-to-head interface is typified by gpU (for gene product U)[6,11] of siphophage lambda and gp17[7,12] of siphophage SPP1. A large number of homologous proteins has been identified in other siphoviruses and myoviruses, indicating that they are an essential component of long tails[6,13]. They were originally named tail terminator proteins[11,14] but recent studies showed that the widespread essential function of this protein superfamily is to act as a tail-to-head joining protein (THJP)[7]. THJPs were reported to be hexamers in the tail structure[6,15] although their precursors during assembly can be monomeric[7,12].

A superfamily of genes encoding a second tail component of the tail-to-head interface in siphoviruses and myoviruses has emerged from bioinformatic analyses. They were named tail completion proteins (TCP)[16] or Ne1[13] (for neck protein of Type 1). TCP genes are found in the genetic

¹Université Paris-Saclay, CEA, CNRS, Institute for Integrative Biology of the Cell (I2BC), 91198 Gif-sur-Yvette, France. ²Université Paris-Saclay, CEA, CNRS, Cryo-Electron Microscopy Facility, Institute for Integrative Biology of the Cell (I2BC), 91198 Gif-sur-Yvette, France. ³Université Paris-Saclay, CNRS, Institut de Chimie des Substances Naturelles, UPR 2301, 91198 Gif-sur-Yvette, France. ⁴Microscopy and Cryo-electron Microscopy Service Group, Max Planck Institute for Molecular Genetics, Ihnestrasse 63-73, 14195 Berlin, Germany. ✉e-mail: isabelle.auzat@i2bc.paris-saclay.fr

**Fig. 1 | Gp16.1 and gp17 in bacteriophage SPP1.**
**a** Composition of the SPP1 phage tail and topology of its protein components (adapted from Auzat et al.[7]). The generic name of proteins engaged on tail assembly are shown at the top of the figure together with their abbreviations. The specific name of the corresponding proteins of phage SPP1 is shown below within brackets (gpX for gene product X). **b** Gene organization of the SPP1 genome region centered on genes *16.1* and *17* that are displayed in cyan and brown, respectively. The upper ruler shows the genome co-ordinates of the SPP1 sequence (accession code X97918.2). The SPP1*sus31* (named SPP1*gp13⁻* from hereafter), SPP1*sus999* (SPP1*gp16.1⁻*) and SPP1*sus82* (SPP1*gp17⁻*) mutations within genes *13, 16.1* and *17*, respectively, are indicated by a black diamond. The precise position and the nature of the mutation are detailed inside the black rectangles. Asterisks indicate stop codons. **c** Composition of purified tails purified from non-permissive infections of different mutants as labeled on top of the gel lanes. The letters "a" and "b" indicate the final step in the tail purification procedure, i.e. glycerol gradient or anion exchange chromatography to remove the last small contaminating protein assemblies, respectively. Tail proteins were separated in a 16% SDS-PAGE and identified by Western blot. Gp19.1 (schematized by a magenta rectangle), gp17 (brown oval) and gp16.1 (cyan oval) were detected with polyclonal antibodies raised against the purified proteins[7,23,27]. Their proposed position in the different tail structures are schematized at the figure bottom.

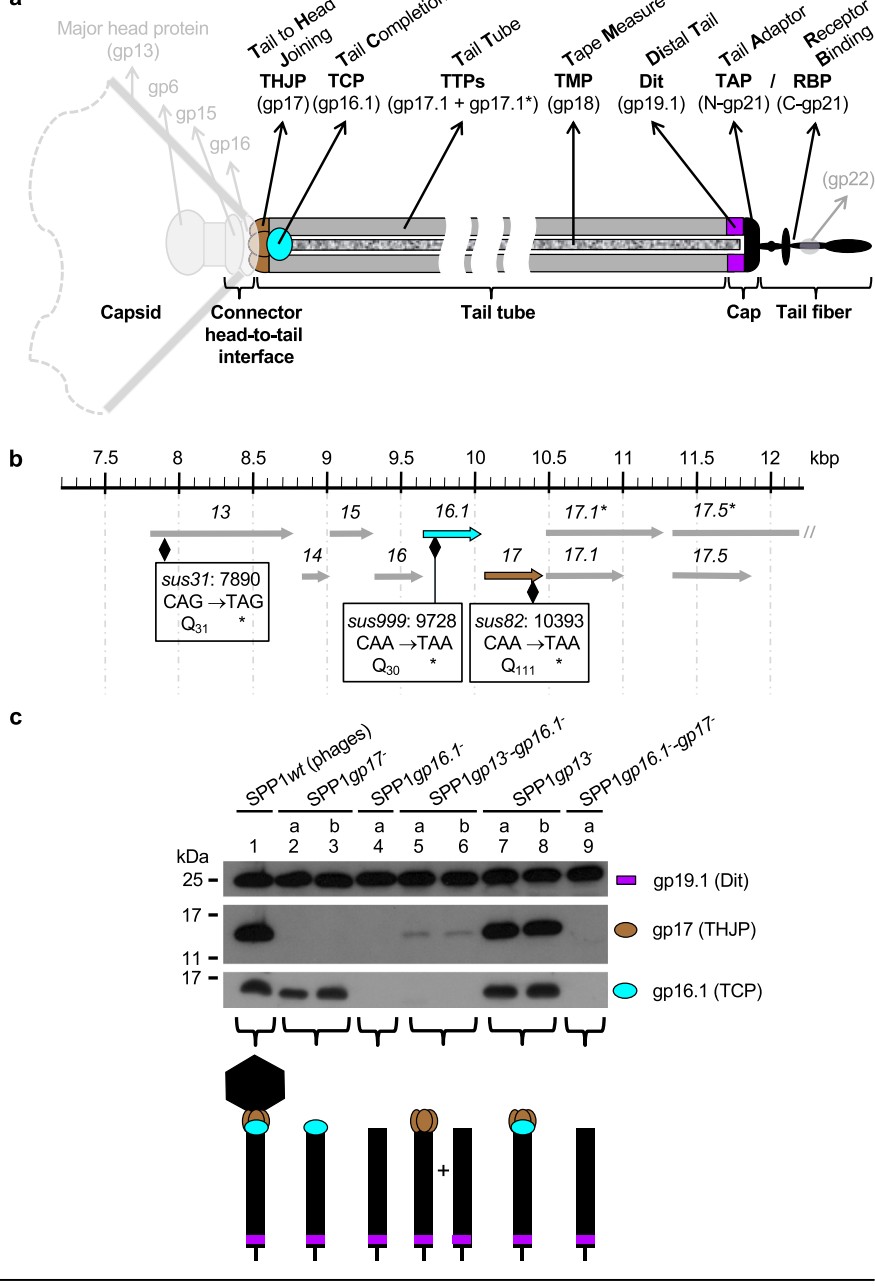

neighborhood of THJP and TTP encoding genes[13] (Fig. 1a, b). The variability of TCP proteins' size and sequence divergence likely prevented earlier assignments to the same protein superfamily, before finding their extensive distribution in phages with long tails[13]. Known TCP-defective mutant phages are not viable[17–21]. Collectively, these findings indicate that TCPs are widespread essential proteins of long tails. In bacteriophage λ, the TCP gpZ was reported to bind to the phage tail structure after the THJP[14]. Phage particles lacking gpZ have a normal morphology but are mostly non-infectious[17,18], a phenotype attributed to misplacement of DNA at the tail-to-head interface preventing its exit through the tail[22]. Electron microscopy (EM) of phage lysates led to the proposal that TCP-defective mutants of myophage Mu also assemble mostly non-infectious phage particles[19] while TCPs of phages P2 and TP901-1 were reported to be required for joining tails to capsids[20,21]. These studies indicate that TCPs may assist assembly of the functional tail-to-head interface and/or ensure DNA correct positioning for delivery to the host. However, they do not establish if TCPs play either one or both of these putative functions in different phages and they did not reveal their underlying mechanisms.

Bacteriophage SPP1 TCP gp16.1 is a component of phage tails. In its absence, tails of normal length are assembled but they lack the THJP gp17 indicating that gp16.1 and gp17 bind sequentially to the tail during termination of tail assembly[23]. Here, we report the purification of SPP1 gp16.1 in a soluble form, overcoming a methodological lock to study TCPs. This achievement provided a framework for rigorous assessment of its role in the phage particle. Gp16.1 assists gp17 binding to phage tails during assembly. However, its intriguing essential function is shown to be to ensure the successful transfer of phage DNA through the bacterial envelope to the bacterial cytoplasm, preventing ejection of the viral genome to the extracellular space.

## Results

### A minor population of SPP1 tails binds stably to gp17 in absence of gp16.1
The use of conditional lethal mutants such as suppressor-sensitive mutants (*sus*), which have a stop codon in the coding region of interest, is a tool of choice for determining function of individual gene products. Mutations *sus*

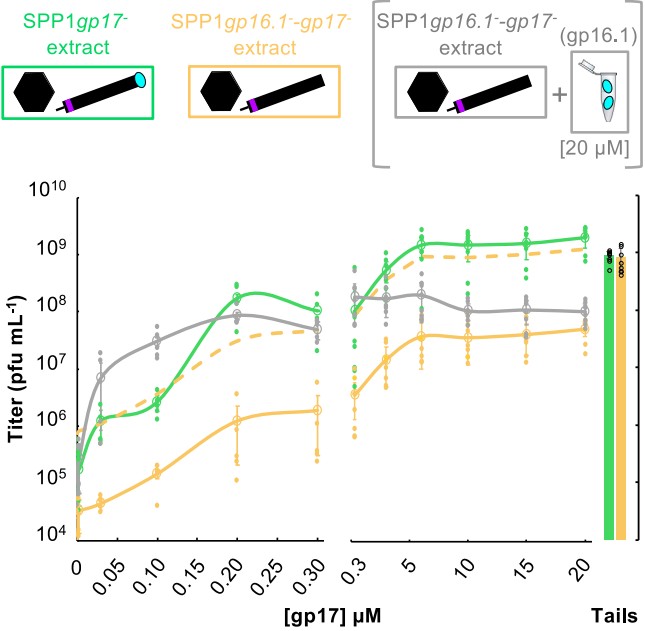

**Fig. 2 | SPP1 assembly in vitro.** The upper part of the figure shows a schematic representation of the phage structures with or without gp16.1 (cyan ovals) found in the two donor extracts used. Extracts from the non-permissive strain *B. subtilis* YB886 infected with the THJP-deficient mutant SPP1*gp17⁻* or with TCP-THJP-deficient mutant SPP1*gp16.1⁻-gp17⁻* are displayed inside green and orange rectangles, respectively. Double mutant SPP1*gp16.1⁻-gp17⁻* extracts mixed with 20 μM purified gp16.1 (cyan ovals in an eppendorf tube) are bracketed in gray. Each extract was mixed separately with increasing concentrations of purified gp17 ranging from 0 (negative control) to 20 μM or with complete tails purified from SPP1*gp13⁻* infected bacteria (positive control). The bottom part of the figure displays counts of infectious particles at the end of the in vitro assembly reactions with added gp17 (curves) or tails (bars). They were scored by titration on the permissive host *B. subtilis* HA101B because the viral particles assembled in vitro were genotypically *sus* mutants. Two datasets, one at low gp17 concentrations (concentration-dependent behavior, left) and another at high gp17 concentrations (plateau behavior; right), were obtained independently for technical feasibility. Curves, bars and experimental points are color coded according to the rectangles and brackets in the figure top. The empty circles correspond to the mean of the experimental points, and the curves connecting them are are an eye guiding support. The hatched orange curve shows titers of SPP1*gp16.1⁻-gp17⁻* multiplied by a 25x factor. Error bars represent the standard deviation.

are suppressed in permissive strains that encode a suppressor tRNA with an anticodon complementary to the stop codon, leading to the insertion of an amino acid in the nascent polypeptide chain when the stop codon is translated[24]. Therefore, infection of a permissive strain allows to multiply bacteriophages carrying stop codons in essential genes. These bacteriophages are able to eject their DNA into the cytoplasm of non-permissive strains, lacking the suppressor tRNA. In such situation the gene with the *sus* mutation is not expressed and the steps of infection affected can be correlated to the absence of its encoded-protein. This approach is used to identify the step(s) of viral assembly that is(are) arrested during non-permissive infection of a phage carrying a *sus* mutation in genes essential for formation of the viral particle[24].

We have shown previously that free tails lack the THJP gp17 when they are purified from bacteria infected with a SPP1 *sus* mutant defective in production of TCP gp16.1[23]. However, the experiment did not exclude the possibility that a sub-population of tails could have bound gp17, in absence of gp16.1, to assemble the tail interface for attachment to DNA-filled capsids present in cells infected with the TCP-defective mutant. If this attachment reaction occurred, tails carrying gp17 could possibly not be detected because they would rapidly mature to complete phage particles. To investigate this hypothesis, we constructed a double SPP1*sus* mutant defective in both

capsid formation and gp16.1 production (SPP1*gp13⁻gp16.1⁻*; see Fig. 1a, b, for gene and protein nomenclature) and analyzed the composition of purified tails assembled during infection of the non-permissive strain *B. subtilis* YB886 (Supplementary Table 1) by this double mutant. Tails assembled in absence of gp17, of gp16.1, of both proteins, and of SPP1 capsids purified from infection with appropriate *sus* mutants[25,26] (this work; Supplementary Table 1; see Methods) were used as controls. The SPP1 tail protein Dit (gp19.1), which forms a hexamer located at the tail extremity distal from the capsid[27] (Fig. 1a), was used to normalize loading for western blots. Gp16.1 was shown to be stably associated to tails assembled in absence of gp17 (Fig. 1c, lanes 2-3). Its band intensity was slightly lower than the one found for phage particles or for complete tails purified from mutant SPP1*gp13⁻* (Fig. 1c, lanes 1 and 7-8, respectively). Infection with SPP1*gp16.1⁻* leads to assembly of tails lacking both gp16.1 and gp17 (Fig. 1c, lane 4), as previously reported[23]. However, when gp16.1 is not produced and capsid assembly is impaired (SPP1*gp13⁻-gp16.1⁻*) there is a small but detectable amount of gp17 associated to purified tails (Fig. 1c, lanes 5-6). Therefore, there is a minor population of tails with stably bound gp17 in absence of gp16.1.

### Role of gp16.1 on the tail-to-head joining reaction
We then investigated the role of gp16.1 to load gp17 into phage tails using an in vitro assembly assay to characterize the tail-to-head joining reaction[7]. Gp16.1 was purified to homogeneity (Supplementary Fig. 1a) and found to have an elution $K_{av}$ (≈0.43) in size exclusion chromatography of a globular protein with an apparent molecular mass of 17.1 kDa (Supplementary Fig. 1b), compatible with the theoretical mass of the 16.5 kDa gp16.1 monomer. The dependence of gp17 concentration to form virus particles was tested in absence of gp16.1 and in presence of either endogenous or of exogenous purified gp16.1. Purified proteins were added to complement in vitro the lysates of non-permissive *B. subtilis* YB886 infected with different SPP1 *sus* mutants (Fig. 2). The first lysate contained DNA-filled capsids and tails with endogenous gp16.1 but lacked gp17 (SPP1*gp17⁻* donor). The second lysate contained DNA-filled capsids and tails but lacked both gp16.1 and gp17 (SPP1 *gp16.1⁻ gp17⁻* donor) (Fig. 1c). We added increasing concentrations of gp17 to the extracts and scored for infectious particles in the permissive strain HA101B to quantify assembly in vitro dependent on gp17. SPP1*gp13⁻* purified tails that carry gp16.1 and gp17 were used as positive control.

When purified gp17 was added to the SPP1*gp17⁻* extract, the number of infectious particles formed raised with the addition of increasing concentrations of gp17 (Fig. 2, green curve). We produced one set of data for low concentrations of gp17 (0.003–0.3 μM: concentration dependent behavior; Fig. 2, bottom left) and another for high concentrations of gp17 (0.3–20 μM: plateau behavior; Fig. 2, bottom right) for technical feasibility. The assembly reaction showed a concentration-dependence on gp17 reaching a plateau at a concentration of 6 μM gp17 (Fig. 2, green curve). Its maximal yield was ~2-fold higher than the one obtained with control purified SPP1*gp13⁻* tails that contain both gp16.1 and gp17 (green bar on the bottom right of Fig. 2). Surprisingly, a similar concentration-dependence behavior was observed when gp17 was added to lysates of the double mutant SPP1*gp16.1⁻-gp17⁻* that does not produce gp16.1 (Fig. 2, orange curve). Therefore, gp16.1 does not seem to favor the concentration-dependence binding of gp17 to tails. However, the titer of infectious phages produced in the in vitro assembly reaction at each concentration of gp17 tested (orange curve) was only ~4% of the one obtained when the tails present in the reaction carry gp16.1 (green curve). This led us to investigate the effect of adding gp16.1 to the assembly reaction of gp17 in the SPP1*gp16.1⁻-gp17⁻* extract. Presence of an excess of purified gp16.1 (20 μM) shifted the gp17 concentration-dependence profile to lower concentrations, in the range between 0.003 to 0.2 μM gp17. A plateau was reached at 0.2 μM gp17 (Fig. 2, bottom, gray curve). At low concentrations of gp17 (0.03 and 0.1 μM), the number of viral particles formed is approximately 5 to 10-fold higher when an excess of purified gp16.1 is added to the in vitro reaction (Fig. 2, gray curve, bottom left) than when gp16.1 is already bound to the assembled tail present in the donor

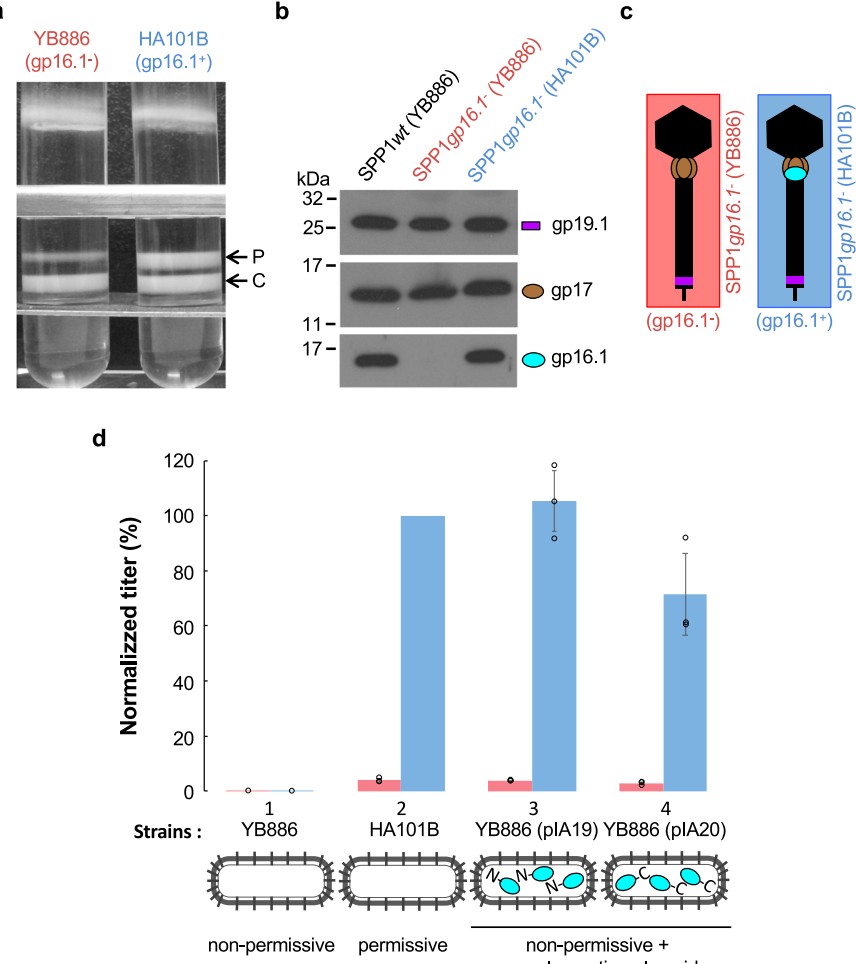

**Fig. 3 | Assembly and infectivity of SPP1gp16.1⁻ virions with or without gp16.1.**
**a** Isopycnic centrifugation of phages produced during SPP1*gp16.1*⁻ infection of YB886 (non-permissive strain) or HA101B (permissive strain). The picture shows tubes after centrifugation of viral particles through a discontinuous density gradient with preformed layers of 1.7, 1.5 and 1.45 g cm⁻³ CsCl in TBT buffer. The upper band (P) corresponds to complete phage particles and the lower band (C) to tailless capsids filled with DNA, as visualized by EM (Supplementary Fig. 2). **b** Presence of gp19.1 (control), gp17 and gp16.1 tail proteins in different CsCl-purified tailed phage particles (upper bands in **a**) determined by Western blot. Symbols are as in Fig. 1c. **c** Schematic representation of phages SPP1*gp16.1*⁻ (gp16.1⁻) (salmon) and SPP1*gp16.1*⁻ (gp16.1⁺) (blue) used in figures from hereafter. They are both genotypically *gp16.1*⁻ but either lack or carry gp16.1, respectively, in the viral particle. This depends on whether phages were produced by infection of strains YB886 or

HA101B, respectively. **d** Infectivity of SPP1*gp16.1*⁻ (gp16.1⁻) and SPP1*gp16.1*⁻ (gp16.1⁺) phage particles. A similar amount of purified physical particles, normalized by DNA content, was titrated in the *B. subtilis* strains schematized underneath the histogram. Gp16.1 encoded by plasmids in YB886 strains is depicted schematically by cyan ovals. The N and C characters show the position of the polyhistidine tag in the protein N-terminal or C-terminal position, respectively. Titers are expressed as a percentage of those obtained for SPP1*gp16.1*⁻ (gp16.1⁺) titrated in HA101B (strain 2) that corresponds to $9.22 \times 10^{11} \pm 1.96 \ 10^{11}$ pfu ml⁻¹. Similar reversion levels were measured in YB886 (strain 1) for gp16.1⁻ and gp16.1⁺ viral particles. They correspond to $7.60 \times 10^{-4} \pm 3.63 \ 10^{-4}$% and $1.03 \times 10^{-3} \pm 4.96 \ 10^{-4}$%, respectively. Bars correspond to the average of the experimental points represented by empty black circles. Results are an average of three independent experiments. Error bars represent the standard deviation.

extract (green curve). We hypothesize that the high concentration of gp16.1 in solution induces a conformational change of gp17 that primes its binding to the tail and its functionalization for viral particle assembly. In contrast, at high concentrations of gp17 the presence of gp16.1 reduces the yield of infectious particles in the assembly reactions (Fig. 2, bottom right).

We conclude that physiological concentrations of gp16.1 in SPP1*gp17*⁻ extracts, where tails were loaded with gp16.1 during infection (green in Fig. 2), does not influence the concentration-dependent binding of gp17 to tails but its absence reduces significantly the number of infectious particles assembled.

**Assembly of SPP1*gp16.1*⁻ phage particles in vivo**
The finding that SPP1 infectious particles are assembled in vitro in absence of gp16.1 prompted us to investigate their formation during infection with the SPP1*gp16.1*⁻ conditional lethal mutant. This mutant was used to infect the non-permissive host *B. subtilis* YB886 and the permissive strain *B. subtilis*

HA101B, as control. Lysates production and viral particles purification were performed in parallel to be strictly comparable. After isopycnic centrifugation in discontinuous CsCl gradients, correctly shaped phage particles were recovered from an upper band (Fig. 3a and Supplementary Fig. 2), that was well separated from a second band containing denser tailless capsids filled with DNA (Fig. 3a and Supplementary Fig. 2). The latter structures are predominant in non-permissive infections (left tube in Fig. 3a) showing that gp16.1 plays a major role in assembly of complete phage particles. Phage particles produced in non-permissive and in permissive infections with SPP1*gp16.1*⁻ have amounts of gp19.1 and gp17 similar to SPP1 wild type particles (Fig. 3b), confirming correct tail assembly. Virions assembled in infections of *B. subtilis* YB886 with SPP1*gp16.1*⁻ lacked gp16.1 (identified by a rectangle with salmon contour in cartoons from hereafter, as in Fig. 3c) while wild type amounts of gp16.1 were found in particles assembled in SPP1*gp16.1*⁻ infections of the permissive strain HA101B (Fig. 3b), as schematized in Fig. 3c (rectangle with blue contour).

Taken together these results show that gp16.1 is important but not essential for assembly of tails competent to bind SPP1 capsids. In its absence, a sub-population of properly shaped viral particles is assembled. We used these purified, correctly assembled, viral particles lacking gp16.1 to study the function of gp16.1 in infection initiation independently of the role of gp16.1 in virus assembly.

## Phage particles lacking gp16.1 are mostly non-infectious

Infectivity of SPP1*gp16.1⁻* phages produced in non-permissive (gp16.1⁻; salmon bars in Fig. 3d) or in permissive conditions (gp16.1⁺; blue bars in Fig. 3d) was compared by phage plaque assays. The total amount of DNA quantified by optical density at 260 nm was used to normalize the number of physical particles present in each population of CsCl-purified phages. Their purity and lack of contaminating tailless DNA-filled capsids was confirmed by EM (Supplementary Fig. 2). Titration of the same number of physical particles showed that absence of gp16.1 in the phage structure correlated with a considerable reduction of phage titer in the permissive strain HA101B when compared to phages carrying gp16.1 (Fig. 3d). Almost 96% of the gp16.1⁻ particles were non-infectious when normalized according to the quantity of DNA. The thermostability of these particles assessed by titration of viable particles (Supplementary Fig. 3a) and by DNA release resulting of particle disruption[28] (Supplementary Fig. 3b) was undistinguishable from wild type SPP1 virions indicating that gp16.1 does not add robustness to the viral particle structure.

SPP1 gp16.1 could exert its function in infection either by controlling DNA ejection from the viral particle or by being ejected into the bacterium to promote host takeover. To distinguish between these two hypotheses, we infected non-permissive bacterial strains producing gp16.1 with phage particles that carry gp16.1 (SPP1*gp16.1⁻* (gp16.1⁺)) or not (SPP1*gp16.1⁻* (gp16.1⁻)). Both infections led to titers similar to the ones found for infection of the permissive strain HA101B showing that gp16.1 encoded by the recombinant plasmid is active to complement the SPP1*gp16.1⁻* mutation. However, presence of gp16.1 in the bacterial cytoplasm before entry of the phage DNA did not restore infectivity of phage particles lacking gp16.1, the percentage of non-infectious phages remaining around 96% (Fig. 3d, strains 3 and 4). The function of gp16.1 in phage particle infectivity is thus conceivably at a stage that precedes entry of SPP1 DNA in the cell cytoplasm and gp16.1 is, most likely, not ejected into the host bacterium.

Considering that only ~4% of phage particles lacking gp16.1 are infectious, we applied this correction factor (x25) to the data obtained in the in vitro assembly experiments of Fig. 2 to score for the total number of particles assembled in absence of gp16.1 (dashed orange curve). Remarkably, the concentration-dependence on gp17 for assembly of total phage particles lacking gp16.1 followed the same pattern as particles containing gp16.1 (Fig. 2). This result shows that, although particle assembly is affected in absence of gp16.1 (Fig. 3a), the major role of gp16.1 is to render the mature phage particles infectious.

The finding that SPP1*gp16.1⁻* is affected at two steps of the SPP1 life cycle prompted the question whether this phenotype results exclusively of the defect in gp16.1 production. We note that SPP1*gp16.1⁻*, carrying a stop codon in gene *16.1*, multiplies normally in the suppressor strain *B. subtilis* HA101B and is complemented in non-permissive strains expressing gene *16.1* (blue bars in Fig. 3d). Therefore, the strong phenotypes observed result of the defect in gp16.1 production. Nevertheless, we sequenced the complete SPP1*gp16.1⁻* genome by NGS to identify potential additional mutations that could contribute to its phenotype. The >25,000-fold sequence coverage allowed detecting robustly mutations present at a frequency of 0.01% in the phage population. No low frequency mutations were found indicating that the phage population is genetically homogeneous. The phage genome carries three single nucleotide substitutions, in addition to the stop codon in gene *16.1*, that are present in the complete SPP1*gp16.1⁻* population when compared to the SPP1 wild type reference sequence (Supplementary Table 2). Two mutations were silent and one led to substitution Val58→Ile in gene *29.1* (Supplementary Table 2) that encodes a protein of unknown function[29]. These mutations are single nucleotide polymorphisms inherited

from the parental phages used to construct SPP1*gp16.1⁻* (Supplementary Table 3). They are found in SPP1*gp16.1⁻* phages amplified in both the permissive (HA101B) and in the non-permissive (YB886) *B. subtilis* strains (Supplementary Table 3). Since the phenotypes found in this study are observed exclusively in non-permissive infections, when gp16.1 is not produced, we conclude that they result specifically of the lack of gp16.1 production.

## Non-infectious SPP1*gp16.1⁻* phage particles eject their DNA normally in vitro

In order to assess if gp16.1 affects SPP1 DNA ejection, we mixed the same number of total phage particles (normalized according to the amount of DNA) of SPP1 wild type, SPP1*gp16.1⁻* (gp16.1⁻) and SPP1*gp16.1⁻* (gp16.1⁺) with the receptor ectodomain YueB780[30]. Incubation at 37 °C triggered DNA ejection in more than 95% of the particles present in the three phage populations (Fig. 4a). EM of SPP1*gp16.1⁻* (gp16.1⁻) and SPP1*gp16.1⁻* (gp16.1⁺) phages incubated with YueB780 at 37°C revealed that their vast majority was empty confirming that they ejected their DNA (Fig. 4b, top). Furthermore, imaging of phage-DNA complexes by adsorption to mica showed that DNA is similarly ejected through the tail tip in both phages (Fig. 4b, bottom).

## Non-infectious SPP1*gp16.1⁻* phage particles eject their DNA outside host bacteria leading to abortive infection

We then investigated if DNA of SPP1*gp16.1⁻* particles, which is ejected through the tail tip "correct" route (Fig. 4), reaches the bacterial cytoplasm at the beginning of infection. To follow the fate of DNA ejected in vivo, we adapted the method of Fernandes et al.[31] to quantify cell-internalized DNA by qPCR. This elegant and effective method eliminates all extracellular phage DNA to quantify only DNA that enters cells. We quantified SPP1*gp16.1⁻* (gp16.1⁻) and SPP1*gp16.1⁻* (gp16.1⁺) DNA delivery to *B. subtilis* YB886 at an input multiplicity (i.m.) of 5 phage particles/bacterium during the first 5 min of infection. At this early stage of infection, no significant replication of viral DNA occurred[32] making it possible to measure the amount of DNA internalized. The maximum amount of cell-internalized SPP1*gp16.1⁻* (gp16.1⁺) DNA was taken as 100% (Fig. 5, bar 2). Only 4.8% of the DNA of particles devoid of gp16.1 reached the bacteria YB886 cytoplasm (Fig. 5, bar 1). This amount was low but significantly above the background level of 0.6% for both phages, determined by incubation of phages with a YB886-derived strain lacking the receptor YueB that is essential for SPP1 infection[33] (Fig. 5, bars 3,4).

The finding that only ~5% of SPP1 particles lacking gp16.1 deliver their DNA successfully to the bacterial cytoplasm led us to image bacteria incubated for 5 min with SPP1*gp16.1⁻* (gp16.1⁺) and SPP1*gp16.1⁻* (gp16.1⁻). EM of strain YB886 mixed with SPP1 phage particles devoid of gp16.1 showed reproducibly the presence of extracellular DNA (highlighted by white dotted oval in Supplementary Fig. 4a, top left panel) whose appearance is similar to purified SPP1 DNA imaged in the same conditions (Supplementary Fig. 4b). When YB886 was mixed with phage particles carrying gp16.1, which leads to productive infections, there was no detectable extracellular DNA (Supplementary Fig. 4a, bottom left panel). Mixing of SPP1 with a control strain that lacks the SPP1 irreversible receptor YueB (YB886 Δ*yueB*) leads to no ejection of DNA to the bacterial interior (Fig. 5) or to the extracellular space (Supplementary Fig. 4a, right panels), as anticipated. We conclude that the DNA ejection process is triggered normally in phage particles lacking gp16.1 when the phage encounters its receptor. However, the ejected DNA fails to enter the cell, remaining in the extracellular space.

## Gp16.1-like tail completion proteins are widespread among long-tailed bacteriophages

Bioinformatics of gp16.1-like tail completion proteins (TCP)[16] or Ne1[13] revealed their presence in numerous siphoviruses and in a group of myoviruses that infect hosts across a large number of bacterial clades[13]. In order to expand those studies we carried out pBlast and phylogeny analyses of

**Fig. 4 | SPP1 DNA ejection in vitro. a** 0.9% agarose gel of DNA protected inside phage particles from DNase digestion before (−) and after (+) DNA ejection. DNA ejection was triggered with 120 nM of purified YueB780 dimers, during 1 h at 37 °C. The virions analyzed are labeled on top of the gel lanes. The quantity of physical phage particles used in each experiment is identical. **b** DNA ejection triggered in vitro by YueB780 monitored by EM of negatively stained samples (top micrographs) and of particles adsorbed to mica (bottom micrographs). SPP1*gp16.1*⁻ (gp16.1⁻, salmon) and SPP1*gp16.1*⁻ (gp16.1⁺, blue) are on the left and of the right of the figure, respectively.

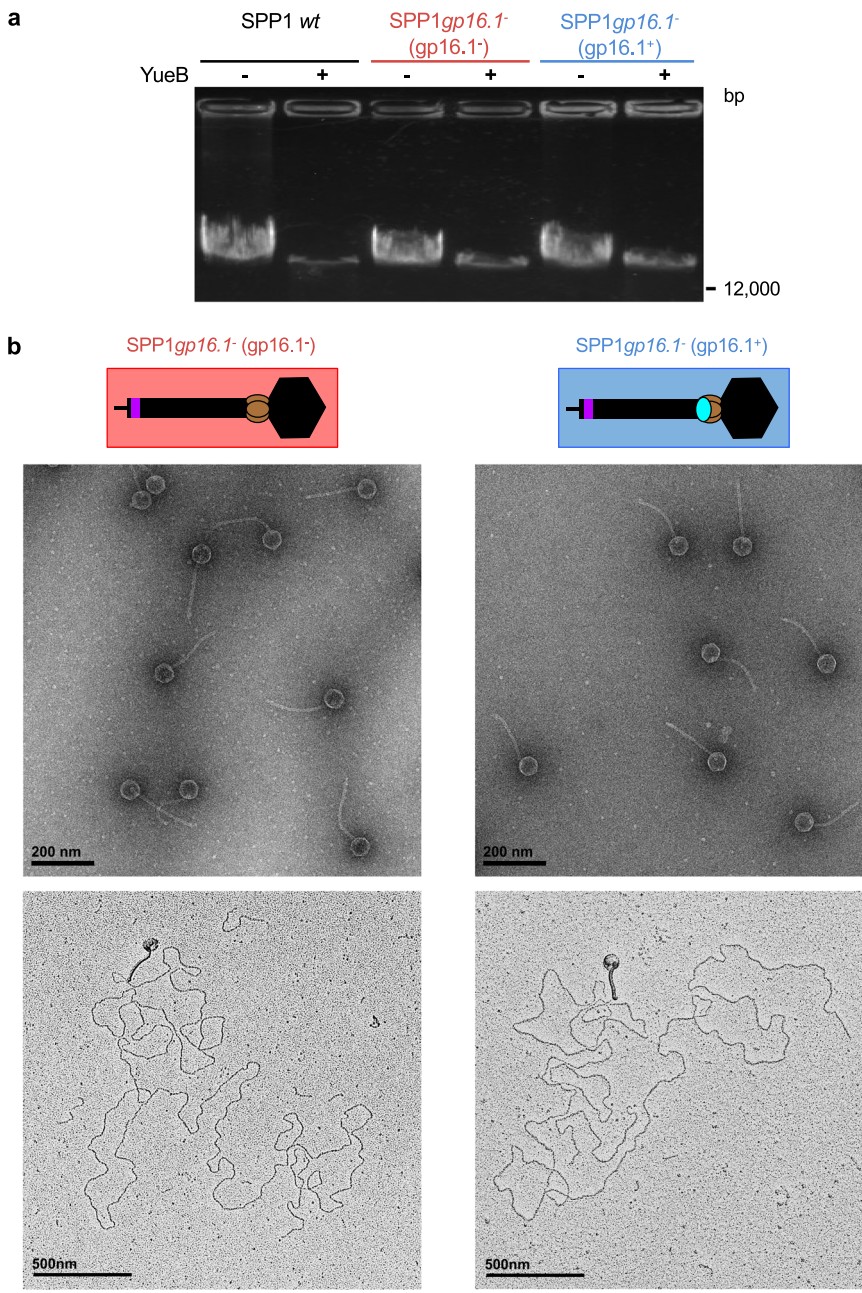

individual TCPs whose function has been investigated experimentally (gp16.1 of phage SPP1[23]; gpZ of lambda[17,34]; p143 of T5[35,36]; ORF40 of TP901-1[21]; gpS of P2[20]; and gpG of Mu[19]) and of the Pfam family 04883 superfamily reference protein gp10 of HK97 (Table 1). The length of these TCPs varies between 112 and 255 amino acids-long. Pairwise alignments revealed protein homology only between TCPs of the two phages infecting Gram-positive bacteria (SPP1 and TP901-1) and of coliphage HK97 while low similarity was detected between the amino termini of coliphages HK97 and T5 TCPs (Supplementary Fig. 5). Their encoding gene precedes the THJP and TTP genes in all five siphoviruses, a feature previously used to identify TCPs[13], but this genome context is lost in myoviruses P2 and Mu (Table 1). Searches with pBlast (e < 10E6 cut-off) in the curated UniProt database showed that all TCPs analyzed have large numbers of homologous proteins (Supplementary Fig. 6, Supplementary Data 1). Each set of homologs contains almost exclusively proteins from phages that infect bacteria from the same taxonomic group (Firmicutes in case of phages SPP1 and TP901-1; Proteobacteria in case of lambda, T5, P2 and Mu).

The noticeable exception is the TCP of phage HK97 that is phylogenetically related to TCPs from phages infecting a broad number of bacterial clades (Supplementary Fig. 6). We have then carried out the same analysis with THJPs of the seven phages, motivated by the cross-talk between the TCP and the THJP identified in this work. Interestingly, the number of THJPs hits was lower than in case of TCPs (see phylogenetic trees in Supplementary Fig. 6, Supplementary Data 2), further highlighting the widespread presence of TCPs in phages with long tails. We propose that TCPs belong to the set of essential proteins necessary for assembly of infectious siphoviruses and of type I myoviruses[13].

## Discussion

A core of conserved proteins builds the backbone of tailed bacteriophage particles. The function of those essential proteins is well documented with the remarkable exception of the "so-called" tail completion proteins (TCPs). The widespread presence of TCPs in siphoviruses and myoviruses was found only recently by combining gene context analysis and sequence gene

homology. Such study led to the identification of genes coding SPP1 gp16.1-like proteins localized between genes of the head completion proteins and the THJP in hundreds of long-tailed phages genomes[13] (Supplementary Fig. 6). TCPs of a few phages were previously shown to be essential but their precise function(s) and mechanisms remained poorly documented. Our work fills this critical gap on tailed phages particles assembly and infection mechanisms, showing that the SPP1 TCP is a tail structural component with two distinct functions. First, it has an auxiliary role on assembly of the tail interface region that binds to the capsid connector and, secondly, it accomplishes a central function on DNA correct routing to the host cell cytoplasm.

Gp16.1 associates stably to the tail structure[23]. The reaction does not require the SPP1 THJP gp17 that achieves the final step of tail attachment to the head connector (Fig. 1c). In contrast, when gp16.1 is absent, correctly-sized tails are assembled but a large majority of those lack gp17 (Fig. 1c). These findings uncovered a first function of gp16.1 to assist association of gp17 to the tail. In this reaction, gp17 monomers assemble a hexamer in the

tail tube end[7] creating the interface of interaction with the SPP1 connector[15]. In absence of gp16.1, there is, however, a minor population of tails with gp17 stably bound (Fig. 1c) that can associate with DNA-filled heads to build complete phage particles. Such particles without gp16.1 are undistinguishable from wild type virions in density, morphology, thermostability, and capacity to eject their DNA in vitro (Figs. 3a, b and 4 and Supplementary Figs. 2 and 3). Gp16.1 is thus important, albeit not essential, to assist loading of gp17 to the tail. Interestingly, tails with or without gp16.1 exhibit a similar gp17 concentration-dependence in in vitro assembly reactions to build viral particles (Fig. 2), suggesting that gp16.1 pre-assembled in the tail is not anymore competent to assist gp17. In contrast, addition of assembly-naïve gp16.1 favors the gp17 assembly reaction (Fig. 2). Collectively, our data support the model that gp16.1, in its free state or rapidly after binding to tail tube end, chaperones gp17 to efficiently interact and form a hexamer at the tail end. The subsequent sturdy attachment of gp17 to the tail connector and the resulting stability of the tail-to-head interface are independent of gp16.1. This mechanism likely explains the requirement of the TCP for formation of a functional head-tail interface of phages P2 and TP901[20,21].

The second and main function of gp16.1 is on infectivity of the mature phage particle. Only ~4% of the physical phage particles lacking gp16.1 are infectious (Fig. 3d). As there are no phage subpopulations carrying compensatory mutations for the gp16.1 defect (see Results), it is likely that stochastically in ~4% of cases the ejected phage DNA successfully crosses the bacterial envelope to reach the bacterial cytoplasm. Low infectivity of TCP-defective lambda and Mu mutants was also reported[14,19]. Our systematic investigation of SPP1 gp16.1 defective particles narrowed down the function of gp16.1 to the correct routing of DNA to the bacterial cytoplasm at the beginning of infection. This happens in spite that DNA ejection triggered by phage encounter with the bacterial receptor occurs, as normally, through the tail end distal from the capsid. However, DNA is released in the extracellular space rather than being delivered across the bacterial envelope to the host cell interior. This raises the fascinating question how TCPs can control delivery of DNA to the bacterium.

An intriguing possibility is that gp16.1 first primes gp17 in the cytoplasm for subsequent interaction with the tail end proximal to the capsid and that, secondly, gp16.1 binds elsewhere in the tail structure to achieve its function for correct routing of phage DNA to the cytoplasm. This hypothesis would provide a biological role to the proposed association of phage T5 putative TCP (p143) to the phage tail fiber protein[36]. In such case, the TCP is positioned for direct interaction with the host cell envelope to ensure accurate traffic of phage DNA to the host cell. However, the finding that gp16.1 binds stably to SPP1 tails in absence of gp17 implies that the TCP is not set to follow obligatorily the order of interactions described above.

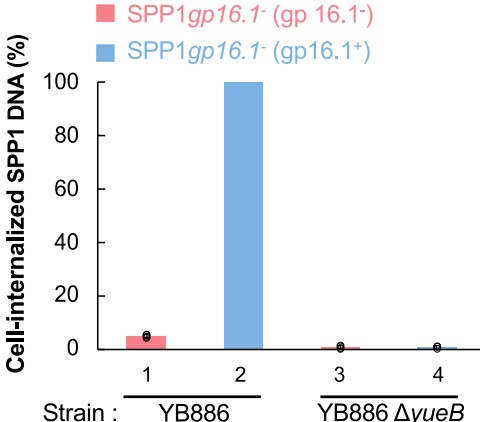

**Fig. 5 | Ejection of phages SPP1*gp16.1⁻* (gp16.1⁻) and SPP1*gp16.1⁻* (gp16.1⁺) DNA in vivo.** Quantification of cell-internalized SPP1 DNA determined by qPCR. Results are presented as percentage of the SPP1 chromosome equivalents internalized after infection of strain YB886 by phage SPP1*gp16.1⁻* (gp16.1⁺) (bar 2). Control experiments were carried out with strain YB886 Δ*yueB* that is defective in the SPP1 bacterial receptor YueB (bars 3 and 4). Bars are an average of four independent experiments represented by empty black circles. Error bars represent standard deviation.

**Table 1 | Characteristics of the TCPs whose function was studied experimentally and the THJPs of the same phages**

| Phage | Siphoviruses | | | | | Myoviruses | |
|---|---|---|---|---|---|---|---|
| | **SPP1** | **Lambda** | **T5** | **TP901-1** | **HK97** | **P2** | **Mu** |
| Tail Completion Protein (TCP) | gp16.1 | gpZ | p143<br>T5.147 | ORF40<br>Tap | gp10 | gpS | gpG<br>Mup31 |
| Uniprot entry | O48447 | P03731 | Q6QGE0 | Q77K21 | Q9MCS9 | P36934 | Q01261 |
| Length (aa) | 141 | 192 | 255 | 112 | 149 | 150 | 156 |
| Experimental evidence for presence in tail | MS - WB | – | MS structure | – | – | – | – |
| Phenotype | Non-infectious<br>Tail-head junction | Non-infectious<br>Tail-head junction | – | Tail-head junction | – | Tail-head junction | Non-infectious |
| TCP's genomic context | THJP −1<br>TTP −2 | THJP −1<br>TTP −2 | THJP −1<br>TTP −2 | THJP −1<br>TTP −2 | THJP −1<br>TTP −2 | THJP +1<br>TTP −10 | THJP −5<br>TTP −8 |
| Tail to Head Joining Protein (THJP) | gp17 | gpU<br>TrP | p142<br>T5.146 | ORF41 | gp11 | gpR | gpK<br>Mup37 |
| Uniprot entry | O48448 | P03732 | Q6QGE1 | Q77K20 | Q9MCS8 | P36933 | Q9T1V8 |
| Length (aa) | 134 | 131 | 161 | 129 | 115 | 155 | 182 |

*MS* mass spectrometry, *WB* western blot, *Tap* tail activator protein, *TTP* tail tube protein, *TrP* terminator protein.

Another conceivable hypothesis is that gp16.1 exerts its two functions at the tail-to head interface, acting as a clamp that reduces the rate of DNA ejection to a level consistent with efficient delivery to the bacterium (Fig. 6). Interaction of the phage with the irreversible receptor on the bacterial surface triggers DNA ejection. However, DNA release needs to be preceded by localized digestion of the bacterial cell wall, likely achieved by the tail tip, and establishment of a lipophilic channel in the bacterial membrane for DNA passage. The tape measure protein (TMP) that occupies the lumen of the phage tail tube (Figs. 1a and 6) is ejected before phage DNA and is expected to form the membrane channel. Successful infection thus relies on precise timing between the phage binding to the receptor, building the path for DNA passage through the cell envelope, and exit of the DNA from the particle into the bacterial cytoplasm. Premature DNA exit would result in aborted ejection due to the release of genetic material outside the target cell. Gp16.1 could straddle the apex of the tail by forming a ternary complex with the TTP and the TMP at the tail to head-interface to control the correct timing of TMP and DNA release from the phage tail. Such gp16.1 positioning would reduce the internal diameter of the tail tube and retain the TMP, effectively slowing down TMP and DNA exit until a continuous hydrophilic channel is established between the tail capsid and the bacterial cytoplasm.

Thus, the noose is finally tightening around the *modus operandi* of the essential TCP whose function(s) and mechanism(s) remained elusive for half a century of tailed bacteriophage research.

## Methods
### Materials
Molecular biology reagents were from Invitrogen (Carlsbad, USA), New England Biolabs (Ipswich, USA), Novagen Merck KGaA (Darmstadt, Germany), Roche Applied Science (Mannheim, Germany) and Stratagene (La Jolla, USA). Oligonucleotides were synthesized at MWG Biotech AG (Ebersberg, Germany).

Bacterial strains, phages and plasmids are listed in Supplementary Table 1.

*Escherichia coli* DH5α was used for all cloning procedures while *E. coli* BL21 (DE3) pLysS or JS218[37] strains were used for overproduction of gp16.1 and gp17[7], respectively. *B. subtilis* YB886[38] was used as non-permissive strain and to bear plasmids coding for different gp16.1 forms and gp17-6His in in vivo and in vitro complementation assays. *B. subtilis* HA101B (sup-3[39]) was the permissive host for bacteriophages SPP1*gp16.1*⁻ (SPP1*sus999*, this work), SPP1*gp16.1*⁻-*gp17*⁻ (SPP1*sus999-sus82*, this work) and SPP1*gp17*⁻ (SPP1*sus82*[7,26]). The SPP1-resistant *B. subtilis* strain CSJ4 that carries a deletion in the phage receptor *yueB* (YB886 Δ*yueB*)[33] was used as a negative control for SPP1 DNA delivery to the bacterial cytoplasm.

### SPP1*sus* mutants construction
SPP1*sus666* (SPP1*gp16.1*⁻-(*gp17**)) carries a nonsense mutation in gene *16.1*[23] but we found that it has an additional nucleotide change in gene *17*

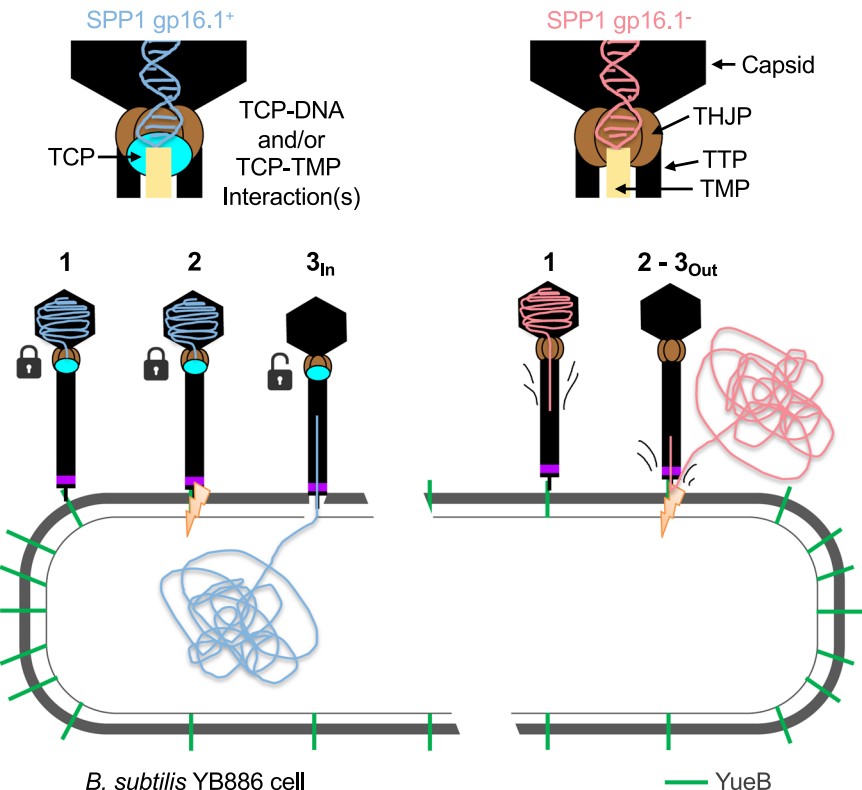

**Fig. 6 | Phage DNA ejection model in the presence or absence of TCP.** The upper part of the figure shows a schematic representation of the interface between the capsid and phage tails with TCP, (wild type phage, left) and without TCP (right). Proteins and the capsid are identified by black arrows. The lower part of the figure shows the steps leading to the ejection of DNA from virus particles: (1) irreversible adsorption to YueB, (2) cell wall localized degradation, (3ₐₙ) DNA passage through the cytoplasmic membrane and entry in the cytoplasm, and (3ₒᵤₜ) DNA ejection into the extracellular space. In the left panel, the closed locks symbolize the action of TCP in delaying DNA ejection by interacting with DNA and/or the TMP (steps 1 and 2). The open lock symbolizes the loss of these interaction(s) leading to DNA ejection (step 3). Without TCP (right panel), DNA and TMP are no longer retained and the DNA slips inside the tail as soon as the phage interacts with the irreversible receptor (step 1). Steps 2 and 3 then become concomitant. The breach in the cell wall not being complete, DNA ejection occurs into the extracellular space.

leading to substitution $P_{23}S$ in gp17. To eliminate this mutation we crossed SPP1sus666 (SPP1gp16.1⁻-(gp17*)) and SPP1sus45[26,40] (SPP1gp17.1⁻). A 3 ml liquid culture of *B. subtilis* permissive strain HA101B culture was infected at $OD_{600} = 0.8$ with the two phages with an input multiplicity of 10 pfu/cfu in the presence of 10 mM $CaCl_2$. After 2 h shaking at 37 °C, cells were harvested and the supernatant was titrated on strain HA101B. 384 isolated lysis plaques were picked and transferred to four 96-well plates filled with 200 μL of TBT buffer (100 mM Tris-HCl, 100 mM NaCl, 10 mM $MgCl_2$, pH 7.5). The double mutant SPP1gp16.1⁻-gp17.1⁻ was screened to not produce virions on YB886 (pIA20) (see *Plasmids construction* section, this work) or in YB886 (pIA23) (this work) that encode 6His-gp16.1 and 6His-gp17.1, respectively, but to multiply on YB886 (pPT25)[40] that encodes gp16.1, gp17 and gp17.1. Two double mutant phages were obtained, one of which lacked the additional mutation on gene *17*. The same phage cross methodology was applied to obtain other SPP1gp16.1⁻ mutants.

In order to obtain the single mutant SPP1sus999 (SPP1gp16.1⁻), we crossed SPP1gp16.1⁻-gp17.1⁻ with SPP1 wild type phages and isolated 288 phage plaques as described above. The desired recombinant phage SPP1sus999 was screened to not produce virions in the non-permissive strain YB886 and to form a lysis spot on strain YB886 (pIA19) (this work) that encodes 6His-gp16.1. Only one positive clone was found. SPP1gp16.1⁻ was then amplified[41] and complete sequencing of a PCR fragment from gene *16.1* to gene *17.1* was carried out at GATC Biotech (Germany). The non-sense mutation in gene *16.1* was present and gene *17* had a wild type sequence. Deep sequencing of the complete SPP1gp16.1⁻ genome was carried out subsequently (see below).

SPP1gp16.1⁻ and SPP1gp13⁻ (SPP1sus31[25]) were crossed to construct SPP1gp13⁻-gp16.1⁻. The double mutant was screened to not produce virions on YB886 (pBT378)[42] or in YB886 (pIA19) that encode gp13 and 6His-gp16.1, respectively, but to multiply in the permissive strain HA101B. Three double mutant phages were obtained from the 288 clones tested.

SPP1gp16.1⁻ and SPP1gp17⁻ (SPP1sus82[7,26]) were crossed to construct SPP1gp16.1⁻-gp17⁻. The desired recombinant phage was screened to not produce virions in YB886 (pIA19) or in YB886 (pIA21)[7] that encode 6His-gp16.1 and 6His-gp17, respectively, but to multiply on YB886 (pPT25)[40] that codes for gp16.1, gp17 and gp17.1. Two double mutant phages were obtained from the 384 clones tested.

### NGS phage genome sequencing
SPP1gp16.1⁻ phage particles, corresponding to 15 μg of DNA, were incubated at room temperature for 30 minutes in TBT buffer containing 50 mM EDTA and RNase at 1 μg.mL⁻¹ in a final volume of 50 μL followed by incubation at 55° for 30 min to disrupt the phages. DNA was extracted twice with 50 μL (1:1) of phenol:chloroform:isoamyl alcohol (25:24:1) and once with 50 μL (1:1) of chloroform. The final upper aqueous phage containing DNA was well dialyzed against 10 mM Tris-HCl, pH 8.0 and stored at 4 °C.

Standard genomic library preparation, Illumina paired end sequencing (2 × 150 bp; approx. 10 million reads) and genome variant detection were performed by Eurofins Genomics.

### Plasmids construction
A 429 bp fragment covering the gene *16.1* sequence (position from 9635 to 10,063 of the SPP1 nucleotide sequence; access code X97918.2) lacking the initiation and stop codons was amplified by PCR from SPP1*wt* DNA with two flanking sequences coding for a 3′-PstI and 5′-AgeI restriction sites. Oligonucleotides F-gp16.1 (5′ ATGCTGCAGgcgcttatgtcggttaga 3′) and R-gp16.1 (5′ TCAACCGGTtcctctcaacctcctcat 3′) were used as forward and reverse primer, respectively. The purified PCR product was completely sequenced at GATC Biotech (Konstanz, Germany), cleaved with PstI and AgeI, and cloned into shuttle vectors pIA2 and pIA3[7], which replicate both in *E. coli* and *B. subtilis* cells, digested by the same restriction enzymes. These plasmids are derived of shuttle vector pHP13[43] engineered with the inducible promoter $P_{N25/O}$[44], a low-copy number plasmid in *B. subtilis* and high copy number in *E. coli*. The resulting plasmids were named pIA19 (6His-gp16.1) and pIA20 (gp16.1-6His).

A 789 bp fragment bearing gene *17.1* was amplified by oligonucleotides F-gp17.1 (5′ ATGCTGCAGccagaaacgccttattatg 3′) and R-gp17.1 (5′ TCA ACCGGTacccgtgctctctgctgg 3′) followed by cloning in the pIA2 vector as described above for gene *16.1*. The resulting plasmid was named pIA23 (6His-gp17.1-17.1*).

### Gp16.1 overproduction and purification
Overproduction of gp16.1 was ineffective in *E. coli* BL21 (DE3) (pIA19) and BL21 (DE3) (pIA20), in spite that the tagged proteins were biologically active in complementation assays when they were produced in *B. subtilis* (Fig. 3d). We tested plasmid pBT361[23], a derivative of vector pRSET-A (Invitrogen), that encodes an N-terminal fusion peptide (36 amino acids long) comprising a polyhistidine tag, a transcript stabilizing sequence from gene *10* of phage T7, the Xpress™ epitope and an enterokinase cleavage recognition sequence. The production of recombinant soluble protein was very good in *E. coli* BL21(DE3) (pBT361) and 5–10-fold higher in the derived strain producing T7 lysozyme encoded by plasmid pLysS. *E. coli* BL21 (DE3) (pLysS) freshly transformed with plasmid pBT361 was grown at 37 °C in LB broth supplemented with ampicillin (100 μg.mL⁻¹) and chloramphenicol (30 μg.mL⁻¹) until reaching an absorbance at 600 nm of 0.5–0.6. Then, cultures were induced with 2 mM IPTG for 3 h. Pelleted bacteria were resuspended in 50 mM Tris-HCl, pH 7.5, 150 mM NaCl supplemented with 1× Protease Inhibitors (Roche) and disrupted by sonication. Crude extracts were clarified by centrifugation at 4 °C twice for 20 min at $27,000 \times g$ and the first step of purification by metal affinity chromatography was performed immediately after cell lysis to minimize the tendency for aggregation of uncleaved protein in solution over time. Recombinant 6His-gp16.1 protein was loaded on a 5 ml Hi-Trap Ni-NTA column (GE Healthcare) pre-equilibrated with buffer A (20 mM $NaH_2PO_4$, pH 7.4, 500 mM NaCl). The column was washed with a step gradient of increasing concentrations of buffer B (buffer A containing 1 M imidazole), and eluted at 250 mM imidazole. Fractions were pooled, diluted to a concentration around 0.5 mg/mL and loaded on a preparative desalting column (HiPrep (26/10) Desalting GE) equilibrated with a buffer optimized for enterokinase cleavage (20 mM Tris pH 7.4, 150 mM NaCl, 2 mM $CaCl_2$). After overnight cleavage of the tag with 0.02 unit/μg enterokinase performed at 16 °C, gp16.1 was run through a size exclusion chromatography column (Superdex 75 HiLoad (26/60) GE) equilibrated with gel filtration buffer (50 mM $NaH_2PO_4$, pH 6.0, 150 mM NaCl). The protein eluted in two peaks. The first eluted in the void volume corresponded to aggregated 6His-gp16.1 while the second contained the gp16.1 cleaved protein. The latter peak fractions were pooled and concentrated by ultrafiltration (Vivaspin 20, 5 kDa molecular mass cut-off). We recovered almost 1 mg of pure protein per liter of culture at the end of the purification process.

### Analytical size exclusion chromatography
Cleaved gp16.1 concentrated at 250 μg.mL⁻¹ was applied to a 24 mL analytical column (Superdex 75 (10/300) GE) equilibrated with gel filtration buffer. The column was calibrated with one vial of protein standards mixture containing thyroglobulin (670 kDa), γ-globulin (158 kDa), ovalbumin (44 kDa), myoglobin (17 kDa) and vitamin B12 (1.35 kDa) (Bio-Rad) (Supplementary Fig. 1b) resuspended in 500 μl of gel filtration buffer. The column void volume ($V_0$) and the total volume ($V_t$) were determined using the elution volumes of blue dextran 2000 and acetone, respectively. The molecular mass of native gp16.1 was estimated by plotting the partition coefficient $K_{av}$ against the log of relative molecular mass for the standards. $K_{av} = (V_e − V_0)/(V_t − V_0)$, where $V_e$ is the elution volume of the protein under analysis.

### SPP1 tails purification
Tail structures were purified using a three-step method from lysates of *B. subtilis* YB886 infected with mutants SPP1sus31 (gp13⁻), SPP1sus82 (gp17⁻), SPP1sus999 (gp16.1⁻), SPP1sus31-999 (gp13⁻-gp16.1⁻), and SPP1sus999-82 (gp16.1⁻-gp17⁻), strictly as described previously[7]. Briefly, large protein complexes were sedimented through a sucrose cushion, the

pellet was resuspended, and tails were partially purified by sedimentation through a glycerol gradient. Elimination of remaining contaminants was performed by chromatography on an anion exchange column[7].

### Preparation of SPP1gp17⁻ and SPP1gp16.1⁻-gp17⁻ extracts from infected cells for in vitro assembly

*B. subtilis* YB886 cultures were infected with SPP1*sus* mutants[42] and aliquots of extracts[7] were incubated with increasing concentrations of purified gp17 protein[7] and gp16.1 in some cases, or with SPP1*gp13*⁻ tails for control. Preparation of extracts and the experimental conditions for in vitro assembly reactions were identical to the ones previously described to assess gp17 activity[7]. Experiments were made in two independent sets corresponding to different ranges of gp17 concentrations due to the complexity of the experimental setup. Two independent experiments with ≥2 technical replicates for each dataset were carried out for low concentrations of gp17 (Fig. 2, left) and four independent biological experiments with 2 technical replicates were carried out for high concentrations of gp17 (Fig. 2, right). Infectious phage progeny was quantified by titration of the assembly reactions final supernatant with the permissive host *B. subtilis* HA101B.

### Production of SPP1gp16.1⁻ phage particles in vivo with different gp16.1 forms

Production of phages lacking gp16.1 or carrying endogenous gp16.1 was carried out by infection with SPP1*gp16.1*⁻ of *B. subtilis* non-permissive YB886 and permissive HA101B strains, respectively. After infection, phage particles were sedimented from phage lysates by overnight centrifugation and run through a discontinuous CsCl density gradient with preformed layers of 1.7, 1.5 and 1.45 g cm⁻³ CsCl in TBT buffer[40,45]. Complete phage particles were recovered from the visible gradient upper band, concentrated and dialyzed against TBT buffer.

### Thermal stability of phage particles

The same number of total phage particles (normalized according to the amount of DNA) of SPP1*gp16.1*⁻ (gp16.1⁻) and SPP1*gp16.1*⁻ (gp16.1⁺) were incubated at different temperatures in a PCR machine with a hot lid, in blocks preheated to the target temperature. Incubations were carried out for 15 min. After cooling for 10 min on ice, the samples were titrated on the permissive strain HA101B.

For Thermal Shift Assay experiments, SPP1 phage preparations were diluted in buffer 100 mM Tris-HCl, pH 7.5, 100 mM NaCl containing either 1, 2.5, 10 mM MgCl$_2$ or 10 mM EDTA. SYBR gold was diluted 3000-fold from a 10,000-fold stock solution (Invitrogen). Reaction mixtures were made in a 96-well fast PCR plate with $5 \times 10^7$ particles per reaction at a final volume of 20 µl. The temperature gradient was carried out in the range of 10 °C to 99 °C at 3 °C/min with a StepOnePlus real-time PCR system (Applied Biosystems). Fluorescence was recorded as a function of temperature in real time (excitation with a blue light-emitting diode (LED) source and emission filtered through a Joe emission filter). The capsid disruption temperature, assessed by the release of DNA, was calculated with StepOne software v2.2 as the maximum of the derivative of the resulting SYBR gold fluorescence curves.

### DNA ejection in vitro

For DNA ejection in vitro assay, the concentration of CsCl-purified SPP1*wt*, SPP1*gp16.1*⁻ (gp16.1⁻) or SPP1*gp16.1*⁻ (gp16.1⁺) phage particles was normalized by A$_{260 \text{ nm}}$ according to their DNA content. $3 \times 10^9$ particles in each ejection reaction were incubated with 5 units of Benzonase and 10 µg of RNAse for 30 min at 37 °C in ejection buffer (100 mM Tris-HCl, pH 7.5, 300 mM NaCl, 10 mM MgCl$_2$). Ejection was triggered by the addition of the receptor (2.8 µg of YueB dimer), or H$_2$O for control, in a final volume of 15 µL. The receptor/phage molar ratio in the mixtures was ~340 to ensure an excess of receptor. Samples were supplemented with 25 units of Benzonase to reduce viscosity and genome ejection allowed to proceed for 1 h at 37 °C. Samples were deproteinized[30] and protected DNA was analyzed in a 0.9% agarose gel (~10⁹ phage particles of each ejection reaction were loaded).

For EM, approximately $7 \times 10^{10}$ phage particles were incubated 30 min on ice with 0.1 µg of YueB dimers and 25 units of Benzonase in ejection buffer in a final volume of 10 µL. DNA ejection was triggered by incubation at 37° for 1 hour. Aliquots of 2 µL samples were used for negative staining with 2% uranyl acetate[46]. The same protocol with 400-fold diluted phages and without Benzonase was used for EM observation after adsorption to mica[47].

### DNA ejection monitored in vivo by qPCR

Overnight cultures of *B. subtilis* YB886 and YB886 Δ*yueB*[31,33] strains were diluted 1:100 in fresh LB medium and grown at 37 °C to OD$_{600}$ of 0.8. Six aliquots of 1 mL of each strain were taken, centrifuged at 5000 × *g* for 5 min and resuspended in 1/10 volume of LB medium supplemented with 10 mM CaCl$_2$. Half of the aliquots of each strain were infected at an input multiplicity of 5 with SPP1*gp16.1*⁻ (gp16.1⁻) phages, the other half with SPP1*gp16.1*⁻ (gp16.1⁺) phages for 5 min. In order to quantify only the phage DNA internalized in bacteria, free input DNA-filled phages, reversibly or irreversibly associated with the host cell envelope without (or partially) transferring their DNA to the bacterial cytoplasm, were disrupted by high Temperature-EDTA-Benzonase (TEB) treatment[31]. This treatment quickly killed bacteria without causing significant cell lysis. Briefly, after 5 min of infection, samples were rapidly diluted 20-fold with disrupting buffer (100 mM Tris-HCl, pH 7.5, 100 mM NaCl, 50 mM EDTA), vigorously vortexed for 5 seconds and incubated for 15 min at 65 °C. Samples were centrifugated for 10 min at 14,000 × *g*, resuspended in 100 µL of digesting buffer (100 mM Tris–HCl, pH 7.5, 100 mM NaCl, 100 mM MgCl$_2$) supplemented by 0.25 U/µL of Benzonase and incubated 1 hour at 37 °C.

Cells were then washed twice with inactivation buffer (100 mM Tris-HCl, pH 7.5, 50 mM EDTA), the supernatant was completely removed and 500 µl of resuspension buffer (50 mM Tris-HCl, pH 8.0, 50 mM glucose, 50 mM NaCl, 2 mg.mL⁻¹ lysozyme and Protease Inhibitors 1x (Roche)) were added to the pellets to lyse cells for 10 minutes at room temperature. Cell extracts were treated with 500 µl of lysis buffer (50 mM Tris-HCl, pH 8.0, 50 mM NaCl, 1% NP40, 10 mM EDTA, 100 µg.mL⁻¹ RNAse) for 30 min on ice. Then, proteins were eliminated with 0.5% SDS and 50 µg.mL⁻¹ proteinase K for 1 h at 65 °C and DNA was stocked at −20 °C until the qPCR experiments.

Quantitative real-time PCR (qPCR) was performed on a QuantStudio 12 K Flex Real-Time PCR System (Life Technologies) with a SYBR green detection protocol according to the manufacturer's instructions. DNA extracts were first diluted 300-fold with H$_2$O and 3 µl of diluted samples were mixed with Fast SYBR Green Master Mix and 500 nM of each primer in a final volume of 10 µL. The reaction mixture was loaded into 384-well microplates and submitted to 40 cycles of PCR (95 °C, 20 s; [95 °C, 1 s; 60 °C, 20 s] ×40) followed by a fusion cycle to analyze melting curves of PCR products. All the qPCR reactions were made in technical duplicates. Primers were designed using the Primer-Blast tool from NCBI and the Primer Express 3.0 software (Life Technologies) to quantify SPP1 gene 6 (G6-Forward: 5' CGGGCTGAAATACCTGTGGA 3' and G6-Reverse: 5' TAGCCCCTCCTCCGATTGTT 3') and *B. subtilis* gene *gyrA* (GyrA-Forward: 5' GAATACGGCAGAACGGCAAA 3' and GyrA-Reverse: 5' TTCGTTTTGAAACCCCATGC 3'). Specificity and the absence of multilocus matching at the primer site was verified by BLAST analysis. The amplification efficiencies of primers were determined using the slopes of standard curves obtained over a five-fold dilution series. Amplification specificity for each real-time PCR reaction was confirmed by analysis of dissociation curves. qPCR technical duplicate measurements were made for each of the 4 biological replicates. Determined Ct values were then used for further analysis. The ratio numbers of SPP1 phages per *B. subtilis* genome were determined using the ΔΔCt method[48].

### DNA ejection monitored in vivo by EM

Bacterial strains were grown, aliquots of 2 mL were taken at OD$_{600}$ of 0.8, centrifuged at 10,000 × *g* for 3 min and resuspended in 1/10 volume of LB preheated at 37 °C and supplemented with 10 mM CaCl$_2$. Tubes were

incubated for 3–4 min at 37 °C in a static water bath and aliquots of 25 μL of each strain were infected with SPP1*gp16.1⁻* (gp16.1⁻) or SPP1*gp16.1⁻* (gp16.1⁺) for 5 min at an input multiplicity of 60. 3 μL of the phages-bacteria mix were rapidly transferred to a glow-discharged grid and LB culture medium was completely removed by 3 successive passages of the grid in 100 μl of washing buffer (50 mM Hepes, pH 7.5, 50 mM NaCl, 10 mM MgCl₂). The grid was transferred to a 100 μL drop of washing buffer containing glutaraldehyde to a final concentration of 1%. Fixation was carried out for 5 min at room temperature (20 °C). After 3 passages of the grid in 100 μL of washing buffer, the grid was neutralized in glycine buffer (50 mM glycine, pH 7.8, 100 mM NaCl, 10 mM MgCl₂) for 30 min at room temperature. Finally, the grid was washed once, stained with a low concentration of uranyl acetate (0.2%) to minimize over-staining of bacteria. The grids were visualized at 100 kV with a Tecnai 12 Spirit transmission electron microscope (Thermo Fisher, New York NY, USA) equipped with a K2 Base 4k x 4k camera (Gatan, Pleasanton CA, USA).

## Bioinformatics

Pairwise protein sequence alignments were made with Protein Blast using default parameters.

Protein BLAST of experimentally studied TCPs and of their corresponding THJPs was performed on the UniProt website[49] using the blastp 2.12.0+ program using the UniProtKB/Swiss-Prot non-redundant protein sequence database. Advanced parameters: E-Threshold = 0.00001, Matrix = Auto-BLOSUM62, Filter = None, Gapped = Yes, Hits = 250 were applied. The TCP and THJP sequences submitted and the graphical presentation of TCP homologs are shown in Supplementary Fig. 6 for each bacteriophage analyzed. The TCP and THJP homologs found for each bacteriophage are presented in Excel tables in Supplementary Data 1 and 2, respectively.

The Protein Blast datasets were used to build phylogeny trees on the NGphylogeny website. A fully automatic workflow[50] including the following modules: input data (Fasta format), multiple alignment (MAFFT), alignment curation (BMGE), tree Inference (FastME) and Tree Rendering (Newick Display) was performed with default parameters. The TCP and THJP trees for each bacteriophage are shown side by side in Supplementary Fig. 6.

## Statistics and reproducibility

The number of independent experimental replicates is indicated for each experiment in the figure legends, except for Fig. 2 where this information is given in the Methods section. Means and standard deviations were determined using MS Excel or GraphPad Prism v10.2.2 for Supplementary Fig. 3b.

## Reporting summary

Further information on research design is available in the Nature Portfolio Reporting Summary linked to this article.

## Data availability

Uncropped images are available in Supplementary Fig. 7. Source data for graphs are provided in Supplementary Data 3. The SPP1 genome sequence was previously deposited in GenBank under accession code X97918.2. All other data are available from the corresponding author on reasonable request.

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

## Acknowledgements
Naima Nhiri is acknowledged for her most helpful support on qPCR and TSA experiments. Jörg Burger is acknowledged for negative staining of phage preparations used for mica adsorption experiments. We thank Carlos São-José for bacterial strains. This work was supported by institutional funding from CNRS. This work benefited from the CryoEM platform of I2BC, supported by the French Infrastructure for Integrated Structural Biology (FRISBI) [ANR-10-INSB-05-05] and member of IBISA.

## Author contributions
I.A. designed and supervised the research. I.A. carried out molecular biology, biochemistry, bacteriology experiments and bacteriophage handling. M.O. made negative staining of samples, optimization and EM observation. B.F. performed adsorption of phage-DNA complexes to mica and EM experiments with P.T. and T.M. E.J. performed and analyzed qPCR and TSA experiments. I.A. analyzed the data, prepared the figures and wrote the paper with P.T. All authors revised and approved the final version of the manuscript.

## Competing interests
The authors declare no competing interests.
