## [Peer Review File · Communications Biology]

Reviewers' comments:

Reviewer #1 (Remarks to the Author):

The work is fundamental in nature, well designed and conclusions supported. But it complements previous findings and this sense, hinders novelty. After clarification of some concerns stated below, I would recommend this work for publication.

Reviewer #2 (Remarks to the Author):

Dual function of a highly conserved bacteriophage tail completion protein essential for bacteriophage infectivity" by Dr Auzat and colleagues [COMMSBIO-23-3176]

This manuscript characterizes bacteriophage SPP1's tail completion protein (TCP), gp16.1. TCPs belong to a large superfamily of bacteriophage tail proteins, but their fundamental role in bacteriophage assembly and DNA delivery during infection is not well understood. Here the authors show that there are two functions for gp16.1. First, gp16.1 assists assembly of gp17, which is the head to tail joining protein, onto the tail prior to binding to the SPP1 head. Second, when gp16.1 is absent, DNA injection into hosts fails. This latter result is very interesting as the gp16.1 protein is located near the capsid head and not at the tip of the tail that interacts with the bacterium. The authors present very nice and convincing experiments, but the paper is difficult to read and needs a careful editing. Some examples:

I had to look up ref 24 to understand these sentences starting with line 86:

"We have shown previously that tails assembled in infections with a SPP1 mutant defective in TCP gp16.1 production lack the THJP gp17 24. However, the experiment did not exclude that a subpopulation of tails with gp17 could have bound to the connector of DNA-filled capsids."

From ref 24, I now understand that the tails were assembled in the absence of heads. But I still don't understand how tails made as described "defective in TCP gp16.1 production lack the THJP gp17" can now have gp17? The last sentence of the paragraph clarifies this somewhat and perhaps should be moved up. But why did the authors not see this result in the work published in ref 24?

This entire paragraph is nearly unintelligible. I had to look up what a sus mutant is. I'm afraid the use of amber suppressor strains is not common anymore, so the authors should consider adding a sentence explaining amber suppression on mutants, and the amber suppressor and non-suppressor bacterial strains.

Though the authors give the info in the figure 1 legend, it was too hard to keep all of the acronyms straight, so I made my own table of the acronyms and proteins. The authors might consider doing that rather than the long list in the figure legend.

Figure 1, the gels are a bit dark. Thank you for the lovely diagrams throughout the paper.

Line 105: remove “have”

Line 112: we complemented different lysates of SPP1 mutants-infected bacteria, should perhaps be “we complemented lysates generated by infection of SPP1 mutants with purified proteins”.

Figure 2—the smooth curves connecting the data points is weird. Maybe just show the data points without the curves?

Figures 3 and 4 (and others)? Could you just say permissive and non-permissive rather than the strain names? And do the same for the rest of the figures?

Line 147: prompted “us” to investigate...

On many occasions, the closing parenthesis is missing. See lines 310, 318, 334 for examples.

A table of the bacteria and phage strains genotypes/phenotypes in the methods would be helpful.

Love figure 6!

Supplementary figure 4— what is circled with the dashed white line?

Reviewer #3 (Remarks to the Author):

Review for COMMSBIO-23-3176

The manuscript “Dual function of a highly conserved bacteriophage tail completion protein essential for bacteriophage infectivity, by Auzat et al. describes two functions of a tail completion protein in phage SPP1. This work is highly interesting and uses a combination of elegant genetics, biochemistry and electron microscopy. Long tailed dsDNA containing phages often have two proteins in the tail: a tail completion protein (TCP) and a tail-to-head joining protein (THJP). Previous work has indicated that these TCPs may assist during assembly at the head/tail junction, and/or that they might ensure proper DNA delivery to the host. This work reports on the role of proteins gp16.1 and gp17, in bacteriophage SPP1 infection, and is a leap forward made possible by the ability of the authors to successfully purify gp16.1. The main conclusions are that gp16.1 has two functions: 1) minor role to assist gp17 assembly into the tail and 2) to direct DNA ejection productively into the host. Without gp16.1 most of the particles are non-infectious and the DNA spews out of the phage into extra-cellular space.

Overall, the manuscript is lovely. Very interesting, well cited, and easy to read overall. Phage genetics experiments can be tricky to describe well, and I applaud the authors for the schematics in each of the images. It really helps to clarify what is shown. The data overall is excellent quality, good statistics are presented and the micrographs are largely very beautiful. I only have a few comments to strengthen what is already an excellent manuscript.

MAJOR

1. The authors cite in general terms that TCPs and THJPs are homologous in related phages. I would like some more information about how gp16.1 and gp17 in SPP1 relate to other phages. Is this likely a conserved trait among siphophages, or narrow to SPP1 close relatives?
2. Can the authors speculate on how they imagine the 4% of particles lacking gp16.1 are still infectious? Given the model for infection that the authors present in Figure 6 is it just random that the spewing went into the cell?

MINOR

1. Define “gp” in the introduction to make it easier for non-phage experts to follow this terminology
2. Line 112: what is “(cf. Fig 1)”? I do not know the “cf” abbreviation.
3. Figure 1c. I am confused why two different methods of purification are needed (“a” and “b” designations for glycerol gradient versus anion exchange). Clarify the text to explain this.
4. Line 226 and supplementary figure 4a: the DNA in the white dotted oval is very hard to see. It might just be the quality of the images submitted for review, but non-EM experts might not see it clearly.

Reviewers' comments:

Reviewer #1 (Remarks to the Author):

The work is fundamental in nature, well designed and conclusions supported. But it complements previous findings and this sense, hinders novelty. After clarification of some concerns stated below, I would recommend this work for publication

We are glad that the reviewer enjoyed our work.

Reviewer 1, Query 1 (R1Q1): This fundamental study investigates the function of a TCP during assembly/infection, using the *Bacillus subtilis* SPP1 as a model. Previous reports (lambda, Mu) already proved that the TCP-defective mutants retain viral particle morphology, and hint the possible role of helping DNA injection into the host cells. Others, including the authors, show that TCP help the assembly of accessory proteins. In this since work is not entirely novel, but the fundamental science behind this work seems to be solid and well designed with logical steps made to elucidate/prove earlier hypothesis i.e. the dual role of TCPs.

Reviewer 1, Answer 1 (R1A1): We respectfully disagree with the reviewer on the novelty of our work. Previous research on gp16.1 TCP-like proteins, with the exception of lambda TCP gpZ, was based exclusively on EM observation of particles present in mutants extracts¹⁹⁻²¹ or mass spectrometry to show presence of the TCP in phage T5 virions³⁵. Those studies provided no indication how this large group of proteins work. Lambda gpZ was shown to participate in the last steps of tail assembly¹⁴ and that some phage non-infectious particles without gpZ are assembled¹⁷⁻¹⁸. The hypothesis proposed to explain no infectivity of phages lacking gpZ was that an incorrect positioning of the DNA at the head-tail interface impaired its exit from the capsid to pass through the tail for delivery to the host bacterium²². Here we show that this is not the case in TCP-free SPP1 particles that are able to eject their DNA *in vitro* as efficiently as wild-type phages. Collectively previous studies on TCPs identified steps in the phage life cycle that require the TCP but provided no mechanistic information how TCPs work.

The work presented here is novel and provides a molecular basis to interpret diverse phenotypes described in the past 40 years of phage research. We characterize the molecular function of a TCP and, most importantly, (i) demonstrate its two precise roles in phage tail assembly and infection, and (ii) narrow down its essential function to DNA delivery to the bacterial cytoplasm after triggering DNA release from the viral capsid.

R1Q2: The language is sometimes difficult to read, and should be improved if possible. For instance, how can one phage infect a non-permission host? "Sus" nomenclature is also used but is difficult to understand or at least makes reading more difficult. These expressions are often used.

R1A2: This type of experiments and language were frequent in the past of phage and bacterial research but we agree that the use of conditional lethal mutants and their genetics becomes less frequent. We explain it didactically on the first paragraph of Results (lines 88-98) of the revised manuscript:

"The use of conditional lethal mutants such as suppressor-sensitive mutants (*sus*), which have a stop codon in the coding region of interest, is a tool of choice for determining function of individual gene products. Mutations *sus* are suppressed in permissive strains that encode a suppressor tRNA with an anticodon complementary to the stop codon, leading to the insertion of an amino acid in the nascent polypeptide chain when the stop codon is translated. Therefore, infection of a permissive strain allows to multiply bacteriophages carrying stop codons in essential genes. These bacteriophages are able to eject their DNA into the cytoplasm of non-permissive strains, lacking the suppressor tRNA. In such situation the gene with the *sus* mutation is not expressed and the steps of infection affected can be

correlated to the absence of its encoded-protein. This approach is used to identify the step(s) of viral assembly that is(are) arrested during non-permissive infection of a phage carrying a *sus* mutation in genes essential for formation of the viral particle.”

R1Q3: A major concern is how phage mutants were generated, by mixing of phage variants. The identify the right mutants was assessed by sequencing only a small part of the genome (some genes). How can authors cope with the possibility that other mutation might accumulated in the phage genome that could interfere with the interpretation of the results? There is also a possibility of having a mixture of phage population (with and without mutations) present in the same solution. This would be possible to determine with deep sequencing to understand if there is alternative base pairs in the mutations made. If present, could explain for instance in the presence of the minor population of SPP1 tails binds stably to gp17 in “absence” of gp16. Or that “only ~4% of phage particles lacking gp16.1 are infectious”.

R1A3: The issue raised by the referee is interesting and we followed her/his suggestion to deep-sequence the SPP1*gp16*⁻ genome. We note, however, that the phenotypes observed result specifically of mutation SPP1*gp16*⁻ for several reasons:

First, after crossing mutant phages, single clones are isolated and re-isolated by a second passage on plates before phage amplification. This method warrants amplification of pure clonal lines. Phages are amplified in permissive conditions that warrant production of gp16.1 in the cell during phage infection. This avoids pressure to select potential second site mutations compensating for lack of gp16.1. In case such mutations would arise to account for the 4% infectious particles in the SPP1*gp16*⁻ population this would lead to a similar reversion of ~4% when titrating the phage suspension in a non-permissive strain. The reversion of SPP1*gp16*⁻ is much lower (Figure 3d). Secondly, SPP1*gp16*⁻ is complemented by a plasmid coding for the complete gene *16.1* sequence. This demonstrates that the single suppressor sensitive mutation in the phage genome is the stop codon in gene *16.1*. Thirdly, in all studies *in vivo* we compared the phenotypes of infection in conditions that gp16.1 is produced (permissive conditions) or not produced (non-permissive conditions) using the same SPP1*gp16*⁻ preparation to carry out such infection of the two different host strains. Phenotypes are thus specifically associated to gp16.1 presence vs absence.

The entire SPP1*gp16*⁻ genome was deep-sequenced to establish the complete genetic background of this phage mutant and to guarantee the absence of mutations occurring during phage amplification. We found 3 mutations in its genome when compared to our SPP1 wild type reference strain. Those mutations are present in ~100% of the sequence reads. No low frequency mutations were detected (deep-sequencing sensitivity: > 0.01 % of the phage population genomes). Further sequencing showed that the 3 mutations in SPP1*gp16*⁻ originate from its parental phages (Supplementary Table 3). The mutations are present in the original SPP1*gp16*⁻ phage stock as well as in the SPP1*gp16*⁻ phage preparations with and without gp16.1 that were produced in parallel infections for studies carried out in this work. This analysis is described in a new paragraph of the revised manuscript version (lines 217-235).

R1Q4: I would also recommend to make a brief bioinformatic analysis to demonstrate how concerned this dual-function TCP protein is, across phage taxonomy/hosts, which would increase the impact of this work.

R1A4: Our previous collaborative bioinformatics work (reference 14) showed that SPP1 gp16.1 TCPs (named Ne1 in reference 14) are widespread in siphoviruses and in a large group of myoviruses. The study in reference 14 made this assignment based on protein sequence alignments, HHPred and gene context because TCP/Ne1 protein sequences are too divergent for reliable identification of homologs between distantly related long-tailed phages. This study is described both in the manuscript Introduction (lines 60-65) and in Discussion (lines 305-310) addressing reviewer’s 1 and 3 (R3Q1) queries. To further

document the bioinformatics analysis we carried out pBlast using known TCPs identified according to phenotype (phages SPP1, lambda, TP901, P2 and Mu) and/or presence in the phage tail combined with gene context (phage T5). This identified their homologs in hundreds of phages and established their phylogenetic relationship as now reported in the last section of Results (lines 276-299), Table 1 and Supplementary Figures 5 and 6.

Other comments

R1Q5: L19-28: Abstract: It is clearly from the text what are the roles of TCP. This information, also emphasized in the title, should be clearly indicated (as latter is in lines 247-248).

R1A5: We thank the reviewer for the suggestion to provide a more clear-cut description in the Abstract of the two functions of the SPP1 TCP. The text was revised to enhance the findings reported in this manuscript.

“Infection of bacteria by phages is a complex multi-step process that includes specific recognition of the host cell, creation of a temporary breach in the host envelope, and ejection of viral DNA into the bacterial cytoplasm. These steps must be perfectly regulated to ensure efficient infection. Here we report the dual function of the tail completion protein gp16.1 of bacteriophage SPP1. First, gp16.1 has an auxiliary role in assembly of the tail interface that binds to the capsid connector. Second, gp16.1 is necessary to ensure correct routing of phage DNA to the bacterial cytoplasm. Viral particles assembled without gp16.1 are indistinguishable from wild-type virions and eject DNA normally *in vitro*. However, they release their DNA to the extracellular space upon interaction with the host bacterium. The study shows that a highly conserved tail completion protein has distinct functions at two essential steps of the virus life cycle in long-tailed phages.”

R1Q6: L65: please insert reference

R1A6: Done

R1Q7: L148: Mutant was used to infect the non-permissive host? How can a phage infect a host that does not allow replication (non-permissive)? This leads to the question, if gp16.1 aids DNA injection into the host, how can this SPP1gp16.1- generate viral particles? Was the DNA electroporated inside the cells?

R1A7: Amplification of SPP1*gp16.1* is carried out in the permissive HA101B strain. This means that the viral particles produced contain the gp16.1 protein, but their genome has the *16.1* mutation. These particles are therefore able to infect both permissive and non-permissive strains (see R1A2). Once the non-permissive strain has been infected, the assembly and/or infectivity of new viral particles will be affected according to the specific role of the protein no longer produced in such infection conditions. The rationale of these experimental strategies are now described in the first section of Results (lines 88-98) to render comprehensive the use of suppressor-sensitive mutants for assessing gene function throughout the manuscript.

R1Q8: L159: “while wild type amounts of gp16.1 were found in particles assembled in “SPP1gp16.1-infections of the permissive strain HA101B”. How that a protein gp16.1 be generated if not encoded by the phage? Was the host complemented with this protein? Language is not clear enough to allow interpretation.

R1A8: This wording is explained in the new paragraph at the beginning of Results (lines 88-98) that describes the rationale of *sus* mutants experiments to investigate gp16.1 functions. Please refer also to R1A2 and R1A7.

R1Q9: L168: here and in many other places is not quickly perceptible how phages could infect non-permissive hosts. Other language should be used to better clarify.

R1A9: Please refer to R1A2 and R1A7.

R1Q10: L218: Significant lower? Where are the standard deviations or statistical analysis to support this claim?

R1A10: Standard deviation bars are provided in the histogram of Fig. 5. The error bars in the figure show that background values of the experiment, determined in the strain lacking the phage receptor, (bars 3 and 4) are significantly lower than in the strain infected with SPP1*gp16.1*⁻ (gp16.1⁻) (bar 1).

R1Q11: L310: “)” missing

R1A11: we thank the referee for spotting this omission: corrected!

R1Q12: L331: better rephrase sentence by saying “not to produce” instead of to “produce neither”

R1A12: the sentence was modified to (lines 409-410):

“The double mutant was screened to not produce virions on YB886 (pBT378) or in YB886 (pIA19) that encode gp13 and 6His-gp16.1, respectively...”

R1Q13: L347: were these shuttle vectors with high replication rate; inducible?

R1A13: We added the sentence (lines 436-438):

“These plasmids are derived of shuttle vector pHP13⁴³ engineered with the inducible promoter P_{N25/0}⁴⁴, a low-copy number plasmid in *B. subtilis* and high copy number in *E. coli*.”

R1Q14: L392:398: This reviewer would like to see more details to allow reproducibility of this method.

R1A14: We used exactly the same purification method as described in detail in ref 7 (Auzat et al. 2014). We have modified the text accordingly (lines 491-496):

“Tail structures were purified using a three-step method from lysates of *B. subtilis* YB886 infected with mutants SPP1*sus31*⁻ (gp13⁻), SPP1*sus82*⁻ (gp17⁻), SPP1*sus999*⁻ (gp16.1⁻), SPP1*sus31-999*⁻ (gp13-gp16.1⁻), SPP1*sus999-82*⁻ (gp16.1-gp17⁻) as strictly described previously⁷. Briefly, large protein complexes were sedimented through a sucrose cushion, the pellet was resuspended, and tails were partially purified by sedimentation through a glycerol gradient. Elimination of remaining contaminants was performed by fractionation on an anion exchange column⁷.”

R1Q15: L413: “no” typo ?

R1A15: It is no typo but we changed the sentence for clarity (line 512):

“Production of phages lacking gp16.1 or carrying endogenous gp16.1 was carried out by infection with SPP1*gp16.1*⁻ of *B. subtilis* non-permissive YB886 and permissive HA101B strains, respectively.”

Reviewer #2 (Remarks to the Author):

Dual function of a highly conserved bacteriophage tail completion protein essential for bacteriophage infectivity" by Dr Auzat and colleagues [COMMSBIO-23-3176]

This manuscript characterizes bacteriophage SPP1's tail completion protein (TCP), gp16.1. TCPs belong to a large superfamily of bacteriophage tail proteins, but their fundamental role in bacteriophage assembly and DNA delivery during infection is not well understood. Here the authors show that there are two functions for gp16.1. First, gp16.1 assists assembly of gp17, which is the head to tail joining protein, onto the tail prior to binding to the SPP1 head. Second, when gp16.1 is absent, DNA injection into hosts fails. This latter result is very interesting as the gp16.1 protein is located near the capsid head and not at the tip of the tail that interacts with the bacterium. The authors present very nice and convincing experiments, but the paper is difficult to read and needs a careful editing. Some examples:

We are happy that the referee enjoyed our work. The manuscript was revised according to her/his suggestions as detailed below.

R2Q1: I had to look up ref 24 to understand these sentences starting with line 86:

"We have shown previously that tails assembled in infections with a SPP1 mutant defective in TCP gp16.1 production lack the THJP gp17 24. However, the experiment did not exclude that a subpopulation of tails with gp17 could have bound to the connector of DNA-filled capsids."

From ref 24, I now understand that the tails were assembled in the absence of heads. But I still don't understand how tails made as described "defective in TCP gp16.1 production lack the THJP gp17" can now have gp17? The last sentence of the paragraph clarifies this somewhat and perhaps should be moved up. But why did the authors not see this result in the work published in ref 24?

R2A1: The addition of recombinant gp17 protein to bacterial extracts producing tails without TCP and capsids is necessary and sufficient to form phages (Fig 2 and Auzat *et al.* 2014). Therefore, we posited that if a sub-population of tails has stably bound gp17 it could attach rapidly to DNA-filled capsids that are present in TCP-defective infections, as in the work of ref 24, to form phage particles. If present, those phage particles would precipitate in the pellet of the glycerol gradient during tail purification and would not be detected in the experiments of ref 24. In such case, only tails without gp17 would accumulate during infection with the TCP-defective mutant. In order to determine whether tails with gp17 can assemble in absence of gp16.1 and to trap them, we constructed a double mutant that produces neither gp16.1 nor the major capsid protein for this work (SPP1gp13gp16.1⁻). We have now further detailed in the manuscript why free tails with gp17 and without gp16.1 were not detected in the experiment by Seul *et al.* in lines 103-106:

"We have shown previously that free tails lack the THJP gp17 when they are purified from bacteria infected with a SPP1 *sus* mutant defective in production of TCP gp16.1²³. However, the experiment did not exclude the possibility that a sub-population of tails could have bound gp17, in absence of gp16.1, to assemble the tail interface for attachment to DNA-filled capsids present in cells infected with the TCP-defective mutant. **If this attachment reaction occurred, tails carrying gp17 could possibly not be detected because they would rapidly mature to complete phage particles.** To investigate this hypothesis, we constructed a double SPP1*sus* mutant defective in both capsid formation and gp16.1 production (SPP1gp13gp16.1⁻; see Fig. 1a, b for gene and protein nomenclature) and analyzed the composition of purified tails assembled during infection of the non-permissive strain *B. subtilis* YB886 (Supplementary Table 1) by this double mutant."

R2Q2: This entire paragraph is nearly unintelligible. I had to look up what a *sus* mutant is. I'm afraid the use of amber suppressor strains is not common anymore, so the authors should consider adding a

sentence explaining amber suppression on mutants, and the amber suppressor and non-suppressor bacterial strains.

R2A2: The paragraph was revised to improve clarity (lines 99-108) and a new paragraph was added at the beginning of Results to explain the rationale of experiments using *sus* mutants (lines 88-98) (see also R1A2 and R1A7).

R2Q3: Though the authors give the info in the figure 1 legend, it was too hard to keep all of the acronyms straight, so I made my own table of the acronyms and proteins. The authors might consider doing that rather than the long list in the figure legend.

R2A3: We agree with the referee's remark and thank him/her for it. However, no list or table of abbreviations/definitions is permitted by the journal. Therefore, we added the protein names just above their acronyms at the top of figure 1a and removed them from the legend.

R2Q4: Figure 1, the gels are a bit dark. Thank you for the lovely diagrams throughout the paper.

R2A4: To comply with Nature Research's image integrity policies, we have decided to not modify the contrast of the western blots images.

R2Q5: Line 105: remove “have”

R2A5: We thank the referee for this suggestion: done! (line 122)

R2Q6: Line 112: we complemented different lysates of SPP1 mutants-infected bacteria, should perhaps be “we complemented lysates generated by infection of SPP1 mutants with purified proteins”.

R2A6: We revised the sentence for the sake of clarity (lines 128-129):

“Purified proteins were added to complement *in vitro* the lysates of non-permissive *B. subtilis* YB886 infected with different SPP1 *sus* mutants (Fig. 2)”

R2Q7: Figure 2 the smooth curves connecting the data points is weird. Maybe just show the data points without the curves?

R2A7: We understand the referee's remark. However, we think that the curves connecting the experimental points are an useful eye guiding support for easy follow-up by readers. It is now stated in the legend of figure 2 that curves are an eye guiding support.

R2Q8: Figures 3 and 4 (and others)? Could you just say permissive and non-permissive rather than the strain names? And do the same for the rest of the figures?

R2A8: The difficulty is that in Figure 3d there are two non-permissive strains, each harbouring a different plasmid, and in Figure 5/Supplementary Figure 4 there are two different non-permissive strains (the non-permissive “wild type” strain and the non-permissive strain lacking the *yueB* gene that codes for the bacterial receptor of SPP1). We thus preferred to maintain the present labels and did not modify those figures. No bacterial strains are used in Figure 4.

R2Q9: Line 147: prompted “us” to investigate...

R2A9: We thank the referee for this suggestion: done! (line 167)

R2Q10: On many occasions, the closing parenthesis is missing. See lines 310, 318, 334 for examples.

R2A10: We thank the referee for spotting it: done!

R2Q11: A table of the bacteria and phage strains genotypes/phenotypes in the methods would be helpful.

R2A11: We thank the reviewer for this suggestion. The new Supplementary table 1 lists the bacterial strains, phages and plasmids used in this work.

R2Q12: Love figure 6!

R2A12: Great that the reviewer liked our cartoon visual support in Fig. 6. We also find that it is very useful to aid understanding the model described in Discussion.

R2Q13: Supplementary figure 4— what is circled with the dashed white line?

R2A13: We thank the reviewer for calling our attention to this point. The white dashed oval encircles positively stained DNA to help general readers visualize it. This information is now provided in Supplementary Fig. 4a legend. See also R3A6 answer to referee #3.

Reviewer #3 (Remarks to the Author):

Review for COMMSBIO-23-3176

The manuscript “Dual function of a highly conserved bacteriophage tail completion protein essential for bacteriophage infectivity, by Auzat et al. describes two functions of a tail completion protein in phage SPP1. This work is highly interesting and uses a combination of elegant genetics, biochemistry and electron microscopy. Long tailed dsDNA containing phages often have two proteins in the tail: a tail completion protein (TCP) and a tail-to-head joining protein (THJP). Previous work has indicated that these TCPs may assist during assembly at the head/tail junction, and/or that they might ensure proper DNA delivery to the host. This work reports on the role of proteins gp16.1 and gp17, in bacteriophage SPP1 infection, and is a leap forward made possible by the ability of the authors to successfully purify gp16.1. The main conclusions are that gp16.1 has two functions: 1) minor role to assist gp17 assembly into the tail and 2) to direct DNA ejection productively into the host. Without gp16.1 most of the particles are non-infectious and the DNA spews out of the phage into extra-cellular space.

Overall, the manuscript is lovely. Very interesting, well cited, and easy to read overall. Phage genetics experiments can be tricky to describe well, and I applaud the authors for the schematics in each of the images. It really helps to clarify what is shown. The data overall is excellent quality, good statistics are presented and the micrographs are largely very beautiful. I only have a few comments to strengthen what is already an excellent manuscript.

... We are glad that the referee enjoyed our work and manuscript.

MAJOR

R3Q1: 1. The authors cite in general terms that TCPs and THJPs are homologous in related phages. I would like some more information about how gp16.1 and gp17 in SPP1 relate to other phages. Is this likely a conserved trait among siphophages, or narrow to SPP1 close relatives?

R3A1: TCPs and THJPs are widespread in phages with long tails. We included a new section at the end of results to further document this point that is also addressed in the answer to reviewer 1 (see R1A4 above).

R3Q2: 2. Can the authors speculate on how they imagine the 4% of particles lacking gp16.1 are still infectious? Given the model for infection that the authors present in Figure 6 is it just random that the spewing went into the cell?

R3A2: Yes, our current hypothesis is that randomly in ~4% of the cases the ejected phage DNA is crosses successfully the bacterial envelope to reach the bacterial cytoplasm. This is not due to a sub-population of phages carrying compensatory mutations for the *gp16.1* defect (see R1A3 above). We also address this issue in the revised Discussion (lines 334-337).

MINOR

R3Q3: 1. Define “gp” in the introduction to make it easier for non-phage experts to follow this terminology

R3A3: We thank the reviewer for spotting this point. The abbreviation is now explained in line 54 and in the legend of Figure 1 legend.

R3Q4: 2. Line 112: what is “(cf. Fig 1)”? I do not know the “cf” abbreviation.

R3A4: “cf.” is a commonly used abbreviation for the Latin word confer, meaning "compare."

R3Q5: 3. Figure 1c. I am confused why two different methods of purification are needed (“a” and “b” designations for glycerol gradient versus anion exchange). Clarify the text to explain this.

R3A5: The tails recovered in the glycerol gradient are not entirely pure, and passage through the anion exchange column removes some contaminating protein structures.

For clarity, we added this information in the figure 1c legend:

“The letters “a” and “b” indicate the final step in the tail purification procedure, i.e. glycerol gradient or anion exchange chromatography to remove the last small contaminating protein assemblies, respectively.”

R3Q6: 4. Line 226 and supplementary figure 4a: the DNA in the white dotted oval is very hard to see. It might just be the quality of the images submitted for review, but non-EM experts might not see it clearly.

R3A6: This is the way DNA comes out in EM of negatively stained samples. Negative staining is not the method of choice to visualize DNA but we did not have another EM method to image simultaneously bacteria, phages and DNA. This is the reason why we show negatively stained purified SPP1 DNA in Supplementary Fig. 4b as an imaging control and also why we orient non-specialist readers with a dashed oval to see the region with DNA on the top left panel of Supplementary Fig. 4a.

Note that Supplementary Fig. 4a provides visual evidence that phages SPP1*gp16.1*(gp16.1) eject their DNA to the extracellular medium but the key quantitative experiment demonstrating that these phages do not eject DNA to the bacterial cytoplasm is shown in Fig. 5.

REVIEWERS' COMMENTS:

Reviewer #1 (Remarks to the Author):

This reviewer congratulates the authors on the effort to elucidate some issues and to provide additional experiments to support the conclusions.

Reviewer #2 (Remarks to the Author):

I am satisfied by the response of the authors to the reviews. I recommend publication.

Reviewer #3 (Remarks to the Author):

The authors have addressed my previous concerns very well. Congratulations on a lovely body of work.

REVIEWERS' COMMENTS:

Reviewer #1 (Remarks to the Author):

This reviewer congratulates the authors on the effort to elucidate some issues and to provide additional experiments to support the conclusions.

Reviewer #2 (Remarks to the Author):

I am satisfied by the response of the authors to the reviews. I recommend publication.

Reviewer #3 (Remarks to the Author):

The authors have addressed my previous concerns very well. Congratulations on a lovely body of work.

We would like to sincerely thank the reviewers for their time, helpful comments and positive opinion on our work and manuscript.